biomechanics/biomedical engineering/mechanical engineering

prosthesis, optimization, amputation

**Author for correspondence:**
Cara Gonzalez Welker
e-mail: cgwelker@stanford.edu

# Shortcomings of human-in-the-loop optimization of an ankle-foot prosthesis emulator: a case series

Cara Gonzalez Welker[1,2], Alexandra S. Voloshina[3], Vincent L. Chiu[2] and Steven H. Collins[2]

[1]Department of Bioengineering, and [2]Department of Mechanical Engineering, Stanford University, Stanford, CA 94305, USA
[3]Department of Mechanical and Aerospace Engineering, University of California Irvine, Irvine, CA 92697, USA

(iD) CGW, 0000-0003-2769-501X; VLC, 0000-0002-4265-7882

Human-in-the-loop optimization allows for individualized device control based on measured human performance. This technique has been used to produce large reductions in energy expenditure during walking with exoskeletons but has not yet been applied to prosthetic devices. In this series of case studies, we applied human-in-the-loop optimization to the control of an active ankle-foot prosthesis used by participants with unilateral transtibial amputation. We optimized the parameters of five control architectures that captured aspects of successful exoskeletons and commercial prostheses, but none resulted in significantly lower metabolic rate than generic control. In one control architecture, we increased the exposure time per condition by a factor of five, but the optimized controller still resulted in higher metabolic rate. Finally, we optimized for self-reported comfort instead of metabolic rate, but the resulting controller was not preferred. There are several reasons why human-in-the-loop optimization may have failed for people with amputation. Control architecture is an unlikely cause given the variety of controllers tested. The lack of effect likely relates to changes in motor adaptation, learning, or objectives in people with amputation. Future work should investigate these potential causes to determine whether human-in-the-loop optimization for prostheses could be successful.

# 1. Introduction

Over 600 000 individuals in the US live with a major lower limb amputation, with this number expected to double by 2050 due

to the increasing prevalence of diabetes and obesity [1]. Individuals with amputation rely on lower-limb prostheses to replace their lost biological limb. However, gait metrics for users of lower-limb prosthesis are typically worse than those for people without impairment. For example, people using lower-limb prostheses tend to walk more slowly [2], fall more frequently [3], and expend more energy to walk at the same speed [4] when compared with unaffected individuals. In addition, people using a prosthesis tend to demonstrate more gait asymmetry, causing an increased rate of joint degeneration, pain and osteoarthritis in their intact limb [5,6]. The combination of all of these factors can lead to limited mobility for individuals with amputation, resulting in secondary health problems, increased medical costs and more reliance on caregivers [7].

Developing more effective lower-limb prostheses has the potential to mitigate some of these problems for people with amputation. Previous work has found that tuning passive prosthesis parameters can have a small effect on metabolic cost [8,9], while other studies have found metabolic cost to be unaffected [10,11]. In the hopes of producing more meaningful changes for people with amputation, a number of active ankle-foot prostheses have been developed [12–17]. Two different approaches in the development of ankle-foot prostheses have had some success in reducing the energy cost of walking. The first is to provide ankle power similar to biological gait. Lack of push-off work in passive prostheses is implicated in higher step-to-step transition losses [18,19], which are correlated with increased metabolic rate [20]. However, some studies have also shown that more prosthesis push-off work does not necessarily reduce collision work [21,22]. A device providing push-off work and emulating biological characteristics of human neuromuscular control has resulted in mixed results with respect to metabolic cost, with some studies finding modest reductions (e.g. 8% in [23]) and other studies finding no significant difference compared with walking with a passive prosthesis [24]. The second approach focuses on balance assistance in the form of ankle inversion or eversion torque in response to changes in the frontal plane centre of mass. Implementing such control in an ankle-foot prosthesis emulator led to a 9% reduction in energy expenditure during walking when compared with a zero gain controller [25], although walking in the participants' prescribed prosthesis was still the most energetically favourable.

The experimental results of active prostheses show some promise, but biomechanical analyses and simulations suggest that active prostheses have more potential to mitigate problems faced by individuals with amputation. The ankle is estimated to provide approximately half of the power needed for healthy human walking [26], much of which is not replaced by passive prostheses. Simulation models suggest that active prostheses have the potential to reduce the metabolic cost of walking significantly below that of unimpaired walking [27]. One possible reason that powered prostheses have not lived up to their potential is because it is still unknown how best to control them, and physiological and neurological differences between users could lead to varying responses to the same device. Previous work has addressed this by hand-tuning control parameters for each subject, but this process is cumbersome and subjective due the high number of parameters that need to be adjusted. Perhaps a control strategy that takes individual gait characteristics into account would allow powered prosthetic devices to better help mitigate problems for individuals with amputation.

Human-in-the-loop optimization (HILO) has been successfully used to determine control parameters for exoskeletons that result in high reductions in metabolic cost [28–32]. This technique involves choosing the parameters of a control architecture based on the human response to changes in control parameters. To do this in a time-efficient manner, HILO uses an optimization strategy that predicts the optimal parameter set over time using a sample of measurements from different control parameter values. Both covariance matrix adaptation evolution strategies (CMA-ES) [30] and Bayesian optimization [31] have been successfully used to determine control parameters that lead to reductions in metabolic cost. In one example, optimized assistance from an exoskeleton worn on one ankle reduced the energy cost of walking for all participants significantly more than a hand-tuned static controller, with a range of improvements of 14% to 42% and an average improvement of 24% [30]. HILO has been used to successfully reduce the energy expenditure of running [32] and inclined walking [30] with an ankle exoskeleton, in addition to reducing the energy expenditure of walking with a hip exoskeleton [31]. It has also led to reductions in muscle activity while walking with an ankle exoskeleton [30]. These studies suggest that user-specific prosthesis control could provide substantial benefits over conventional, hand-tuned devices. However, HILO has not yet been tested to determine the control parameters of powered prostheses.

In this series of case studies, we applied human-in-the-loop optimization to the control of an active ankle-foot prosthesis used by participants with unilateral transtibial amputation. Four different classes of control architecture were tested: (i) a heel stiffness controller that varied the stiffness and damping of the

**Table 1.** An overview of the protocol for each human-in-the-loop optimization case study that attempted to minimize metabolic rate, including the control architecture and device used, the number of parameters (params) in the control architecture, the time spent walking in each control law with a specific parameter set, and the number of control laws per generation (gen). The participants in each case study are also listed, along with their self-selected walking speed and the number of different control laws that were tested during each bout of continuous walking.

| control architecture | device | params (#) | time/control law (mins) | control laws/ gen (#) | gens (#) | N = | participant | speed (m s⁻¹) | control laws/ bout (#) |
|---|---|---|---|---|---|---|---|---|---|
| heel stiffness | 3 d.f. | 2 | 2 | 6 | 4 | 1 | Sub 1 | 1.25 | 2 |
| neuromuscular | 1 d.f. | 3 | 2 | 7 | 4 | 1 | Sub 2 | 1.0 | 7 |
| balance | 3 d.f. | 2 | 2 | 6 | 4 | 2 | Sub 3 | 0.89 | 2 |
| | | | | | | | Sub 4 | 0.67 | 3 |
| time-based torque | 1 d.f. | 5 | 2 | 8 | 6 | 1 | Sub 1 | 1.25 | 3 |
| | | 4 | 2 | 8 | 4 | 2 | Sub 1 | 1.25 | 3 |
| | | | | | | | Sub 5 | 1.25 | 3 |
| | | 4 | 10 | 8 | 4 | 1 | Sub 2 | 1.0 | 2 |

heel of the prosthesis (figure 1*a*), inspired by the observation that damped articulation of the ankle can reduce energy cost compared with rigid prosthetic ankles [33]; (ii) a neuromuscular controller that emulated biological components of the muscle-tendon complex and has been previously used to reduce the metabolic cost of walking with an active prosthesis [34] (figure 2*a*); (iii) a balance controller that provided ankle inversion/eversion torque based on deviation of the user's lateral centre of mass velocity and has been previously used to reduce the metabolic cost of walking with a prosthesis emulator [25] (figure 3*a*); and (iv) a time-based torque controller with similar control architecture to that used to reduce the metabolic cost of walking with exoskeletons (figure 4*a*). In addition, both five-parameter and four-parameter controllers were implemented for the time-based torque control architecture, resulting in a total of five different control architectures tested.

The objective of the majority of the case studies was to minimize user energy expenditure. The optimization protocol for each case study with this objective is outlined in table 1. Five unique participants (S1–S5) were enrolled in these studies, with some individuals completing more than one experiment (table 2). The protocol was similar for all experiments and was based on an optimization protocol successfully used with exoskeletons [30]. However, there was one exception where the participant completed an extended protocol, in which the time spent in each condition was increased by a factor of five, to determine the effects of increased training on adaptation and optimization. Finally, one additional case study optimized the control parameters with the objective of maximizing participant preference instead of minimizing metabolic rate. For all case studies, participants completed separate validation trials after optimization was complete, and performance with the optimized controller was compared with a generic controller and the participant's prescribed prosthesis. Generic control parameters were based on those used in literature with similar controllers, with subject-specific modifications made in each case study. Based on previous success of human-in-the-loop optimization for ankle prostheses, we hypothesized that the optimized control parameters would result in better outcomes (e.g. reduced metabolic cost or increased preference) compared with any chosen generic parameter set.

## 2. Results

### 2.1. Heel stiffness controller

Two parameters dictating the torque trajectory of the prosthesis heel were optimized in the case study of this novel controller: a heel stiffness parameter and a heel work constant that dictated the amount of work provided or dissipated by the heel during the gait cycle over the course of loading and unloading (figure 1*a*). The optimized parameters after four generations of walking were a heel stiffness of 100 N m rad⁻¹ with a heel work constant of 0.4. In validation, the optimized controller was compared with a generic controller (stiffness = 120 N m, work constant = −0.5) chosen to provide 32% heel energy dissipation, which approximates some energy storage and return (ESR) devices with higher energy dissipation [35]. The optimized controller resulted in a 3.5% reduction in metabolic cost

**Figure 1.** (*a*) The heel stiffness control architecture comprised a stiffness and work constant parameter, which dictated the torque profile during loading and unloading of the heel as a function of heel angle. Stiffness was varied between 50 and 130 N m rad$^{-1}$, and the work constant varied from −0.5 to 0.5, with a larger positive work constant resulting in more positive work injected during the gait cycle. (*b*) The resulting torque profiles of the generic controller and the optimized controller are shown as a function of heel angle. (*c*) The average metabolic cost of the optimized controller was 3.5% lower than the generic controller, but both were higher than the participant's prescribed prosthesis.

compared with the generic controller (optimized: 3.57 W kg$^{-1}$, generic: 3.70 W kg$^{-1}$; figure 1*b*). Both controllers led to a higher metabolic cost when compared with walking with the participant's prescribed prosthesis (3.21 W kg$^{-1}$).

## 2.2. Neuromuscular controller

Three parameters in the neuromuscular controller based on previous literature [34] were optimized. Two were parameters within the muscle-tendon complex model: (i) $F_{max}$, which is analogous to maximum muscle isometric force, and (ii) $\epsilon_{ref}$, which acts as a tendon strain multiplier. The third parameter was a feed-forward gain $K_{ff}$, which affected the magnitude of the muscle activation (figure 2*a*). After four generations of walking, the optimized parameters were: $F_{max} = 4261$ N, $K_{ff} = 1.346$, and $\epsilon_{ref} = 0.062$. We compared the optimized controller with a generic controller with parameters previously determined to best mimic biological ankle torque [34] ($F_{max} = 3377$ N, $K_{ff} = 1.22$, and $\epsilon_{ref} = 0.04$). In validation, the optimized controller resulted in a metabolic cost 1.4% higher than the generic controller (optimized: 2.81 W kg$^{-1}$, generic: 2.77 W kg$^{-1}$; figure 2*b*). Both controllers resulted in higher metabolic cost than walking with the participant's prescribed prosthesis (2.49 W kg$^{-1}$).

## 2.3. Balance controller

Two participants (S3 and S4) completed the four generation optimization protocol for the two parameter balance controller based on previous literature [25]. In this control architecture, the prosthesis behaved as a passive spring in plantarflexion and dorsiflexion, while a baseline nominal inversion/eversion torque $\tau_{nom}$ and a gain dictating the magnitude of correction for centre of mass velocity deviations, $K$, were optimized. (figure 3*a*). The optimization for S3 resulted in a $K$ value of 1.24 and a $\tau_{nom}$ value of −0.173, while the optimization for S4 resulted in a $K$ value of 3.97 and a $\tau_{nom}$ value of 3.03. These were compared with a zero gain generic controller where the $K$ value was set to zero, but the optimized $\tau_{nom}$ value was retained. In validation (figure 3*b*), the optimized controller led to a 1.2% decrease in metabolic cost for S3 (optimized = 2.45 W kg$^{-1}$, generic = 2.48 W kg$^{-1}$) and a 9.1% increase in metabolic cost for S4 (optimized = 1.68 W kg$^{-1}$, generic = 1.54 W kg$^{-1}$). The prescribed prosthesis cost for S3 was less than either controller (2.30 W kg$^{-1}$), while the prescribed prosthesis cost for S4 was equivalent to the generic controller (1.54 W kg$^{-1}$).

## 2.4. Time-based torque controller

### 2.4.1. Introduction

The parametrization of this controller was based on a similar controller successfully used to optimize the parameters of ankle exoskeleton torque to reduce metabolic cost [30], with an additional underlying baseline prosthesis stiffness parameter. In choosing the parameters for the generic controller, the stiffness parameter was based on the approximate stiffness of the participant's prescribed prosthesis,

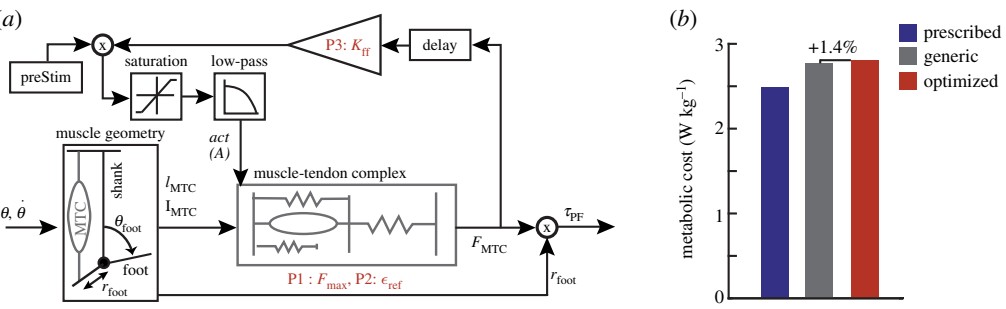

**Figure 2.** (a) The neuromuscular controller draws inspiration from biological muscle-tendon parameters and models the geometry of the prosthetic foot and the biological muscle-tendon complex. Two optimization parameters ($F_{max}$ and $\epsilon_{ref}$) affect the force output from the muscle-tendon complex model, with the third optimization parameter $K_{ff}$ serving as the gain on the force output. Based on biological data, $F_{max}$ was varied from 3000 to 10 000 N, and $\epsilon_{ref}$ from 0.03 to 0.14. Based on prior work, $K_{ff}$ varied from 0.7 to 1.5. (b) The metabolic cost results indicate that the optimized controller was the most costly, followed by the generic controller, and then the prescribed prosthesis. Adapted from: Eilenberg et al. [34].

while the remaining magnitude and timing parameters were based on average optimal parameters for ankle exoskeletons, with subject-specific modifications made for subject comfort.

### 2.4.2. Five-parameter

The five parameters optimized in this control architecture were: baseline stiffness, and the peak time, rise time, fall time, and peak magnitude of the additive torque (figure 4a). Due to the increased number of parameters, the participant completed six generations of optimization before validation instead of four. Optimized parameters were as follows: stiffness = 990 N m rad$^{-1}$, peak time = 48%, rise time = 12%, fall time = 13%, peak torque = −0.01 N m kg$^{-1}$. Note that due to the small magnitude of the optimized peak torque, the optimized controller approximated the behaviour of a passive spring, and therefore varying other timing parameters would have minimal effect on the resulting torque profile. The optimized controller was compared in validation to a generic controller with the following parameters: stiffness = 800 N m rad$^{-1}$, peak time = 86%, rise time = 18%, fall time = 14% and peak torque = 0.05 N m kg$^{-1}$ (figure 4b). The optimized controller resulted in a 3.0% decrease in metabolic cost from the generic controller (optimized = 3.28 W kg$^{-1}$, generic = 3.38 W kg$^{-1}$). However, both controllers resulted in higher metabolic cost compared with walking with the participant's prescribed prosthesis (2.83 W kg$^{-1}$) (figure 4c).

### 2.4.3. Four-parameter

Because the optimized peak torque of the five-parameter time-based torque control optimization had a near-zero magnitude, negating the effect of the other parameters, additional experiments were conducted using a four-parameter time-based torque control architecture with the peak torque set to 0.5 N m kg$^{-1}$. Two participants (S1 and S5) completed the optimization protocol for this control architecture. S1 had previous experience walking with the five-parameter controller, and his resulting optimal control parameters after four generations were: stiffness = 843 N m rad$^{-1}$, peak time = 78%, rise time = 13%, fall time = 21%. Optimal parameters for S5 were: stiffness = 754 N m rad$^{-1}$, peak time = 60%, rise time = 50%, fall time = 17%. Optimized controllers were compared with a generic baseline controller, where stiffness = 900 N m rad$^{-1}$, peak time = 80%, rise time = 10%, fall time = 10% (figure 4b). In validation, the optimized controller resulted in a 5.2% increase in metabolic cost for S1 (optimized = 3.23 W kg$^{-1}$, generic = 3.07 W kg$^{-1}$) and a 0.5% increase in metabolic cost for S5 (optimized = 4.14 W kg$^{-1}$, generic = 4.12 W kg$^{-1}$). For S1, walking with the prescribed prosthesis resulted in a higher metabolic rate than either the generic or optimized conditions (3.40 W kg$^{-1}$). S5 had a lower metabolic rate with his prescribed prosthesis (3.67 W kg$^{-1}$) (figure 4c).

### 2.4.4. Conversationally tuned

During validation of the optimized four-parameter time-based controller, we also tested a fourth condition based on subject preference. To determine controller parameters for this condition, we modified the controller as the participant walked on the prosthesis and provided verbal feedback on

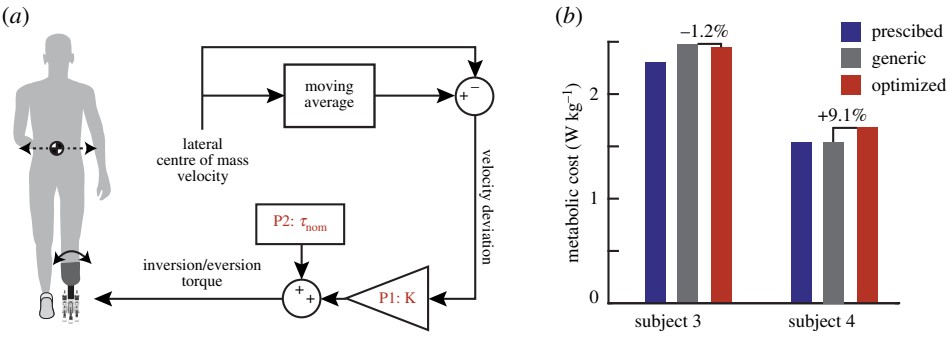

**Figure 3.** (*a*) The balance controller corrects frontal plane deviations of user centre of mass velocity by applying an opposing ankle inversion or eversion torque, scaled by a gain ($K$), in addition to a baseline nominal torque ($\tau_{nom}$). The gain $K$ varied from 0 to 5, and $\tau_{nom}$ varied from −5 to 5 N m. (*b*) The optimized controller resulted in higher metabolic cost compared with the generic controller for both participants, while walking with the prescribed prosthesis led to a user metabolic cost equal to or less than the cost of walking with the generic controller. Adapted from: Kim and Collins [25].

the preferred parameters. The preferred parameters chosen by S1 were: stiffness = 675 N m rad$^{-1}$, peak time = 80%, rise time = 25%, fall time = 15%. The preferred parameters chosen by S5 were: stiffness = 800 N m rad$^{-1}$, peak time = 80%, rise time = 60%, fall time = 20% (figure 4*b*). In validation, the conversationally tuned controllers led to a higher user metabolic cost than all other conditions for both S1 (3.50 W kg$^{-1}$) and S5 (4.36 W kg$^{-1}$) (figure 4*c*). We also collected user opinions of the controllers during validation, without informing the subject which controller they were using. S1 commented that the generic controller was 'comfortable from the start and stayed comfortable', but also noted that it 'has finicky timing'. He noted that both the optimized controller and the conversationally tuned controller were 'comfortable from the start', but the optimized controller 'felt like it required more effort later', while the conversationally tuned controller 'didn't feel good later'. S5 noted that the generic controller had 'too much push-off', the conversationally tuned controller had 'definitely not enough push-off', and the optimized controller had 'a perfect amount of push-off'.

### 2.4.5. Extended protocol

One subject completed an optimization protocol with the four-parameter time-based torque controller in which the duration of each control law was ten minutes instead of the usual 2 min allotted for each condition. The optimization resulted in the following control parameters: stiffness = 710 N m rad$^{-1}$, peak time = 74%, rise time = 44%, fall time = 11%. The optimal controller was compared with a generic controller with the following parameters: stiffness = 1200 N m rad$^{-1}$, peak time = 80%, rise time = 10%, fall time = 10% (figure 4*b*). The optimized controller resulted in an 7.2% increase in metabolic cost compared with the generic controller (optimized = 3.25 W kg$^{-1}$, generic = 3.03 W kg$^{-1}$). Both resulted in higher cost than the participant's prescribed prosthesis (2.61 W kg$^{-1}$; figure 4*c*).

To examine if the additional time spent in each control law affected the estimated steady-state metabolic rate of each condition, we compared the asymptote of the metabolic fit from the first 2 min of data with the average metabolic cost of the last 2 min in each 10 min control law. The RMSE error between these two values was 0.20, or 6% of the participant's metabolic cost while walking with the prescribed prosthesis (standing metabolic cost baseline was not subtracted, as this was collected during optimization and not validation). Because the optimizer uses a weighted average of the control laws resulting in the lowest metabolic cost from each generation to create the distribution of the parameter set for the next generation, we also compared how these rankings differed when calculated using the 2 min metabolic fit versus the average of the last 2 min of each condition. In all generations, the 2 min metabolic fit matched the predictions of the 2 min average for two of the top three control laws that resulted in the lowest metabolic cost. The control law that led to the lowest metabolic cost in each generation was the same for the 2 min fit and the 2 min average 50% of the time.

### 2.4.6. Preference optimization

S1 completed an optimization where the cost function aimed to optimize user comfort, as opposed to minimizing metabolic cost. The protocol was similar to other case studies, with the exception that the

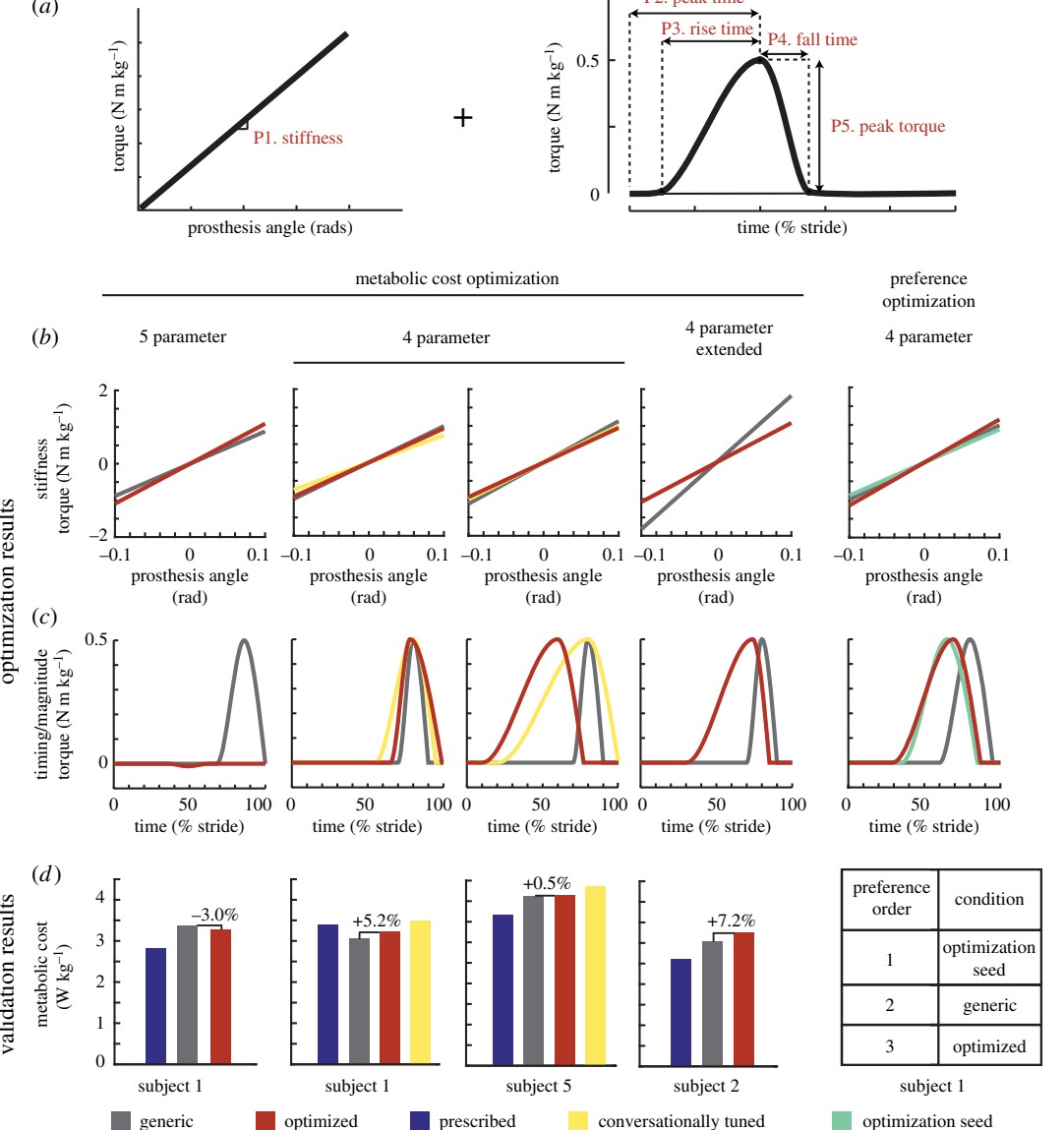

**Figure 4.** (a) The time-based torque controller consisted of an underlying heel stiffness parameter, three timing parameters (peak time, rise time and fall time), and one magnitude parameter (peak torque). Stiffness varied from 400 to 2000 N m rad$^{-1}$, peak time from 10% to 90% of stance, rise time from 10% to 90% of stance, fall time from 10% to 40% of stance and peak torque from −0.5 to 0.5 N m kg$^{-1}$. When using the four-parameter control architecture, peak torque was set at 0.5 N m kg$^{-1}$. (b) Resulting stiffness and torque profiles for each case study using the time-based torque controller. (c) Although the optimized five-parameter controller resulted in a mild metabolic cost reduction, all other optimized controllers resulted in higher metabolic cost than the generic controller. The conversationally tuned controllers resulted in the highest metabolic cost. (d) In the preference optimization, the optimized controller was least preferred in validation.

time spent in each control law was shorter (23 s), and the participant experienced five unique control laws multiple times within a generation for a total of 11 conditions. This allowed all control laws to be directly compared with one another by asking the participant to rank the current control law as better or worse than the previous control law. A composite 'score' for each control law was used at the end of the generation to determine the parameter set of the next generation. At the end of optimization, the parameters were as follows: stiffness = 1041 N m rad$^{-1}$, peak time = 69%, rise time = 40%, fall time = 18%. In validation, in addition to comparing with the generic controller that emulated average optimal magnitude and timing parameters from ankle exoskeletons (stiffness = 900 N m rad$^{-1}$, peak time = 80%, rise time = 20%, fall time = 15%), the initial condition used to seed the optimization was also tested (stiffness = 800 N m rad$^{-1}$, peak time = 65%, rise time = 30%,

fall time = 20%) (figure 4b). During validation, the participant was allowed to request these three controllers in any order, and was asked to rank them at the end of the trial. The participant chose the controller used as the optimization seed as the most preferred, followed by the generic controller (figure 4c). The optimized controller was the least preferred. We also asked for participant feedback on the three controllers. He stated that the optimized controller was 'too passive', and the generic controller 'too springy', while the controller used for the optimization seed was 'just right'.

## 3. Discussion

We present a series of case studies in which human-in-the-loop optimization of an ankle-foot prosthesis failed to produce functionally relevant changes in metabolic cost for people with transtibial amputation. In total, five different control architectures were implemented, and either one or two participants completed the optimization protocol for any one control architecture. Mild reductions in the optimized controller of up to 3.5% compared with a generic controller were seen in some of the case studies. However, because the error in human metabolic rate measurement using indirect calorimetry has been estimated to be between 2 and 3% [36,37], these reductions are not meaningful. In addition, these reductions are significantly smaller when compared with an average 24% reduction in metabolic cost as a result of optimization of unilateral ankle exoskeleton control. In fact, the largest metabolic reductions seen in these case studies compared an optimized controller that behaved as a passive spring with a generic controller that injected mechanical power into the gait cycle, corroborating previous evidence that providing higher mechanical power with an assistive device does not necessarily correlate with reduced user metabolic rate [22,24]. One case study examined an extended protocol, but the optimized controller still resulted in higher metabolic cost than during walking with a generic controller. We also examined a controller chosen by the participant via conversational tuning, but this resulted in higher metabolic rate than all comparison controllers. Finally, in one case study, we attempted to optimize for preference by allowing the participant to rank controllers, but the optimized parameters were not preferred in validation.

Since its inception, human-in-the-loop optimization has been successfully used to optimize behaviour of a range of different exoskeletons for assistance in various gait conditions with different control architectures [28–32]. Given its success with exoskeletons, it is surprising that HILO failed to produce meaningful changes in metabolic cost when used to tune the parameters of an ankle-foot prosthesis controller. There are several reasons why this technique may have failed, which can be broadly separated into two categories. First, it is possible that optimization decisions that have led to successes with exoskeletons are not applicable when optimizing prosthesis control. There are many factors that can affect the optimization, such as the chosen control architecture, optimization protocol, optimization strategy and cost function. It is possible that further modifications of these factors would lead to better results. Second, it is possible that inherent differences in user mechanics and neural control between people with amputation and those without impairment limit the effects of HILO for prosthetic devices. Similarly, the lack of sensory feedback, differences in learning mechanisms, or different objective functions of people with amputation could prevent HILO from being successful.

Although there is a range of decisions that could affect the results of human-in-the-loop optimization, we can hypothesize which are most influential based on the completed case studies. For example, it is possible that the control architectures tested were not capable of reducing the participants' metabolic cost below that of walking with their prescribed prostheses. However, we tested five different control architectures, four of which were vastly different from each other. Both the neuromuscular and the balance controller have previously been successfully used to reduce the metabolic cost of walking with an active prosthesis [23,25]. In addition, a similar time-based torque controller has been used to reduce the metabolic cost of walking with exoskeletons [30]. Given this information, it is unlikely that the controller is the primary reason for the lack of meaningful changes in metabolic cost.

Another factor that could affect the outcome of optimization is the experimental protocol. The protocols chosen for these case studies were based on those successfully used for HILO with exoskeletons with a control architecture similar to the five-parameter time-based torque controller. Past studies showed that for a four-parameter optimization, four generations with eight different conditions of 2 min each resulted in convergence for nine out of 11 participants [30]. In the case studies presented here, the protocol for the four-parameter optimization was the same as previous studies, with additional generations added for additional parameters and fewer control laws tested for

fewer parameters. We also examined an extended protocol, where the amount of time spent in each control law was increased from 2 to 10 min, in order to determine the effect of additional adaptation time and verify that the metabolic fit predicted at 2 min was similar to the average metabolic rate at the end of 10 min. Although there were differences in metabolic rate between the initial 2 min metabolic fit and the metabolic average of the final 2 min, the majority of the top three control laws predicted to result in the lowest metabolic rate at 2 min were the same as those measured at 10 min. Additionally, if a lack of adaptation time was the sole reason that HILO did not succeed, we would still expect to see a consistent reduction in metabolic cost or metabolic cost variability, which was not observed experimentally (see supplemental figures for examples of metabolic rate over the course of the optimization protocol). Nevertheless, it is possible that better results might be achieved with additional training, continued optimization or other protocol improvements.

The optimization algorithm is another factor that could affect the effectiveness of optimization. All case studies presented here relied on a covariance matrix adaptation evolution strategy (CMA-ES) to calculate the next generation of control laws. This strategy has previously been shown to be robust to measurement noise and successful in high-dimensionality spaces. Because CMA-ES does not use information from previous generations, it is also resistant to bias that can occur as a result of adaptation over time. Other optimization strategies such as gradient descent [28,29] or Bayesian optimization [31] have also been effectively used for HILO of exoskeletons to optimize one or two parameters simultaneously. However, gradient descent is sensitive to measurement noise, while Bayesian optimization is not robust to human adaptation over time, as it uses all information from previous trials instead of evaluating each generation independently. It is possible, however, that using a different optimization strategy could be more effective, especially for control architectures with fewer parameters.

Finally, it is important to consider the cost function when defining an optimization problem. In this case, we must take into account two cost functions: the cost function that we choose for the optimizer and the cost function used by the neuromuscular controller of the human during movement. The majority of prior HILO work uses the minimization of metabolic rate as the cost function, both because it is easily quantifiable and because humans walking with exoskeletons have been shown to minimize metabolic rate in real time [38]. However, people with amputation, compared with unaffected individuals, may have additional constraints affecting their gait. For example, individuals with amputation tend to fall more frequently [3] and list socket discomfort as a major limitation [39]. Perhaps because of the additional importance of stability and comfort, people with amputation adapt gait patterns with a similar energy cost independent of prosthesis behaviour. We attempted to address this issue with two case studies: one where we allowed the participant to choose their own device parameters through conversational tuning, and one where we changed the cost function of the optimization to maximize participant preference. However, the conversationally tuned controller resulted in higher metabolic cost than all other controllers tested in validation, and the controller chosen by the optimization for preference was not preferred in validation. This suggests that preference is not highly correlated with energy expenditure for people with amputation. Future work could examine alternative cost functions such as walking speed or stability.

Independent from the optimization variables that can affect the results, it is possible that the contrast between the success of HILO for exoskeletons and the failure of HILO for active prostheses is related to the differences between participant groups. At the joint level, people with amputation lack both sensing and direct control of the joint at which optimization occurs. It is possible that one or both of these characteristics is necessary in order to learn during optimization. For example, when undergoing HILO with exoskeletons, participants are able to feel the timing and magnitude of torque applied by the exoskeleton and can adjust their gait accordingly, either to take advantage of the additional power provided by a 'good' control law or to mitigate the negative effects of a 'bad' control law. By contrast, proprioception in people with amputation is altered, since sensory information is not available directly from the limb. Although people with amputation can fairly precisely sense the stiffness of their prosthesis, sensory information must come from interactions at the socket and whole-body proprioception of resulting knee and hip kinematics [40]. It is possible that additional sensory feedback or direct control would improve adaptation and gait modification, and in turn reduce metabolic cost.

Inherent whole-body level differences between people with amputation and those without impairment could also contribute to the differences in gait adaptation in response to assistive devices. For example, it is possible that individuals with amputation use different learning mechanisms, as neural circuitry associated with motor learning could be re-arranged in complex ways in response to limb loss. Although learning may be possible even with the lack of sensory feedback, it may occur on a much longer timescale than in people without impairment. Specifically, past studies that showed

successful reductions in metabolic cost in response to an active ankle-foot prosthesis have relied on a long adaptation protocol [41]. One limitation of device emulators is that they are constrained to the lab, which limits the length of time and the environment in which participants are allowed to adapt. Both this lack of adaptation time and the differences in prosthesis design could account for the case study results demonstrating that metabolic cost was always lowest with the participants' passive prosthesis. In addition, after limb loss, individuals re-learn how to walk using their passive prosthesis with specific guidance from prosthetists and physical therapists. It is possible that this training interferes with adaptation to a new device. Or perhaps the 'forced exploration' that comes from trying many diverse conditions, which is shown to be beneficial in pushing a person towards the metabolic minimum while walking with exoskeletons [42], is actually harmful to the learning process of people with amputation because 'bad' conditions force the participant to prioritize stability over metabolic cost minimization, which could result in increased metabolic cost.

There are a myriad of reasons why HILO may be more challenging when applied to powered prostheses instead of exoskeletons. Although the results of our case studies showed that HILO does not result in meaningful changes in metabolic cost for people walking with an active ankle-foot prosthesis, it is possible that HILO can be further adapted to be more effective for patient populations. One limitation of our studies is that each experiment contained a limited number of participants and should not be used to make broad conclusions. However, our results can be used to guide future directions for new HILO experiments with active prostheses. Ultimately, HILO has the potential to be extremely beneficial in developing effective control of powered prostheses for people with amputation. Powered prostheses, if controlled properly, could reduce falls, lower user energy expenditure, increase walking speed, and decrease the prevalence of osteoarthritis in the intact limb. We believe the results of our case studies provide a good starting point for future experiments aiming to improve powered prosthesis control through human-in-the-loop optimization.

# 4. Methods

## 4.1. Participants

Five participants with unilateral transtibial amputation ($N = 5$, 4 male and 1 female; age = 37.8 ± 14.1 [26–60] years; body mass = 77.8 ± 8.99 [65.8–90.7] kg; height = 171.2 ± 4.44 [167–178] cm; time since amputation = 13.6 ± 10.4 [3–26] years; mean ± s.d.) took part in the case studies (table 2). An overview of the case studies completed by each participant can be found in table 1. All individuals provided informed consent prior to participation. All study protocols were approved by the Institutional Review Board of either Carnegie Mellon University or Stanford University, depending on the location at which the study took place.

## 4.2. Hardware

In all case studies, participants walked on a treadmill (Bertec, OH, USA) while using either an ankle-foot prosthesis emulator or their prescribed prosthesis. Walking speed for each case study was determined by participant fitness and duration of the study (table 1). The ankle-foot prosthesis emulator consisted of off-board actuation and control hardware attached to a prosthesis end-effector (HumoTech, PA, USA). Flexible Bowden-cable tethers transmitted mechanical power to the prosthesis. Sampling of the strain gauges and encoders of the device, as well as control commands, were implemented at 1000 Hz. All studies used one of two different ankle-foot prosthesis end-effectors (figure 5b). The first was a 1 d.f. device with a mass of 0.96 kg capable of 41° of ankle plantarflexion, 21.7° of ankle dorsiflexion, 190 N m of ankle plantarflexion torque, and 5 N m of ankle dorsiflexion torque (HumoTech, PA, USA). The second end-effector was a 3 d.f. device with an actuated heel and two toe digits and a mass of 1.2 kg. This device could generate 19° of ankle plantarflexion and dorsiflexion, 140 N m of ankle plantarflexion torque, and 100 N m of ankle dorsiflexion torque [43]. The device used for each case study can be found in table 1.

## 4.3. Metabolic rate calculations

To determine user metabolic rate, respirometry data was collected using a Quark CPET metabolic cart (Cosmed, CA, USA). Metabolic rate was calculated using standard empirical equations [44]. In

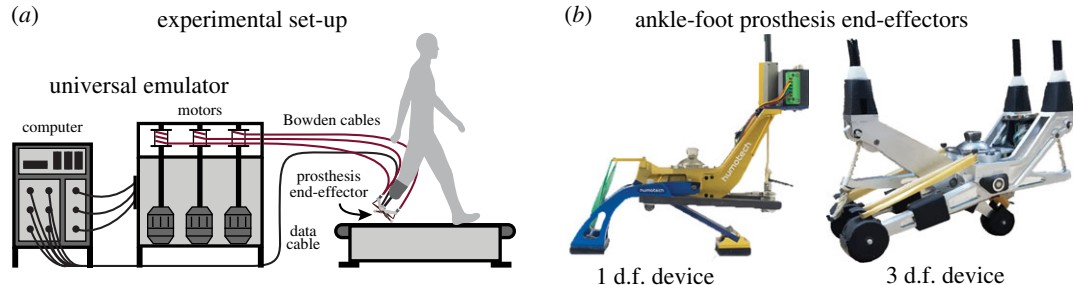

**Figure 5.** (*a*) In all case studies, participants walked on a treadmill in a universal emulator system that consisted of an ankle-foot prosthesis end-effector powered by off-board motors and controlled by a computer. (*b*) Each case study used one of two different ankle-foot prosthesis end-effectors: a 1 d.f. device or a 3 d.f. device.

**Table 2.** Demographics for the five subjects with unilateral transtibial amputation that participated in the case studies. Note that some participants took part in more than one case study, as described in table 1.

| subject | sex | age | body mass (kg) | height (cm) | side of amputation | cause of amputation | time since amputation (years) | prescribed prosthesis |
|---|---|---|---|---|---|---|---|---|
| 1 | M | 26 | 90.7 | 170 | L | Congenital | 26 | Ossur VariFlex XC |
| 2 | M | 26 | 65.8 | 168 | L | Traumatic | 22 | Ossur VariFlex Modular |
| 3 | F | 35 | 74.8 | 167 | L | Traumatic | 3 | Ossur ProFlex LP |
| 4 | M | 60 | 78.0 | 173 | R | Dysvascular | 4 | Ossur ProFlex LP |
| 5 | M | 42 | 79.8 | 178 | R | Traumatic | 13 | Fillauer Wave |

validation, net metabolic rate was calculated by subtracting standing metabolic power from the metabolic cost of all other conditions and normalizing by body mass. During optimization, most control laws were tested for 2 min of walking. As this is not typically sufficient time for metabolic rate to reach a steady state, we fit an exponential curve to the metabolic rate and used the asymptote as the estimate of steady-state metabolic rate, as previously described [30]. One extended protocol tested each control law for 10 min of walking.

## 4.4. Optimization trials

During optimization, user metabolic rate was measured while walking on the treadmill wearing an ankle-foot prosthesis emulator with changing control laws. A covarience matrix adaptation evolution strategy (CMA-ES) was used to identify the control parameters, as described in Zhang *et al*. [30]. The goal of the optimizations was either to minimize metabolic rate or maximize subject preference. In the optimization protocol, a number of control laws are evaluated in each generation, after which the optimizer determines a new distribution from which to select the control laws for the following generation. The number of generations, number of control laws per generation, duration of each control law, walking speed, and the number of control laws tested per continuous walking bout for each case study minimizing metabolic rate is given in table 1. The protocol for the case study in which user preference was optimized is described in further detail in the preference optimization subsection. Prior to starting optimization trials and after every break, participants were allowed to acclimate to walking in the ankle-foot prosthesis emulator controlled using a standard spring controller. The acclimation time was one minute for the case studies using the heel stiffness controller and the four-parameter time-based torque controller with the standard protocol, and 2 min for all other case studies.

## 4.5. Validation trials

At the end of optimization, validation trials were performed in order to compare performance of the optimized controller with one or more of the following conditions: a generic controller, the participant's prescribed prosthesis, and a conversationally tuned controller. The purpose of the comparison between

the generic controller was to compare the optimized parameters to somewhat arbitrarily selected parameters. In order to make these conditions comfortable for the subject, the chosen parameter values for the generic controllers were loosely based on previous literature of similar controllers. Following a standing baseline measurement of energy expenditure, validation conditions were tested twice in a double reversal order to mitigate the effects of adaptation or drift, except the prescribed prosthesis condition was tested only once (with the exception of the balance controller case studies, where it was tested twice). Each validation trial lasted 5 min for the case study with the heel stiffness controller, 10 min for the case study with the extended protocol of the four-parameter time-based torque controller, and 6 min for all other case studies optimizing for metabolic rate. The metabolic rate for each trial was found by averaging the last 2 min of metabolic data. The average metabolic cost of the two validation trials per condition, normalized to user mass, determined the final metabolic cost for that condition.

## 4.6. Control parametrization

### 4.6.1. Heel stiffness controller

This case study used the 3 d.f. ankle-foot prosthesis end-effector. The two toes of the device were connected to passive compression springs, while the heel was the only actively controlled digit. The stiffness of the compression springs was chosen by the participant based on comfort. Heel torque was dictated by a control architecture dependent on stiffness and work constant parameters, as well as heel angle. Heel angle is defined in the sagittal plane as the angle from the plane perpendicular to the prosthesis pylon to the heel end-effector, as described in [43] When the heel angle is less than 0.2 rad, the heel provides a minimum torque of 1.5 N m. As the heel angle increases above this threshold, stiffness and work constant coefficients dictate the heel torque during loading and unloading. The resultant torque profile can be visually displayed as a parallelogram in the heel torque and angle space, with two sides of the parallelogram corresponding to the loading phase of the heel and the other two sides corresponding to the unloading phase (figure 1*a*). The bottom left point of the parallelogram is located at 0.2 rad and 1.5 N m and is the point of transition from minimal torque behaviour at small heel angles to the higher torques at larger angles. The top right point of the parallelogram lies at a heel angle of 0.7 rad, and along a line that begins at the bottom left point of the parallelogram and has a slope equal to the stiffness parameter. This stiffness parameter can vary from 50 to 130 N m rad$^{-1}$.

While stiffness dictates the torque provided at 0.7 rad, the work constant determines the trajectory the torque follows in order to reach this value. Specifically, the work constant affects the top left and bottom right points of the parallelogram, which are defined using a second axis of the parallelogram that transects both points. This second axis is determined such that the heel stiffness does not exceed 130 N m rad$^{-1}$ at any point in the trajectory. Therefore, the top left point of the parallelogram occurs at the point where a line starting at the bottom left of the parallelogram with a slope of 130 N m rad$^{-1}$ intersects a horizontal line with the same torque value that occurs at 0.7 rad (dependent on the stiffness parameter). The bottom right point of the parallelogram occurs at the intersection of a line which originates at the top right point of the parallelogram and has a slope of 130 N m rad$^{-1}$ with a horizontal line at the minimal torque value of 1.5 N m.

After finding the second axis of the parallelogram, the work constant serves as a scaling factor that determines how far along this axis the bottom right and top left points of the parallelogram are located. The work constant is also used to distinguish which points corresponds to the loading and unloading phases. A positive value of the work constant indicates that work is injected during the gait cycle, so the heel produces more torque in the unloading phase as the heel is pushing off than during the loading phase. A negative work constant indicates that the heel produces less torque in the unloading phase, resulting in net negative work over the gait cycle. In addition, the magnitude of the work constant is used to determine the effective 'width' of the parallelogram. A work constant with a value of 0 results in no width, and the torque profile becomes a function of heel angle with a given stiffness for both the loading and unloading phases. A work constant with a higher magnitude results in a greater difference between the torque magnitude in the loading phase and the torque magnitude in the unloading phase.

### 4.6.2. Neuromuscular controller

The neuromuscular controller adapted the active plantarflexor component of the controller described in Eilenberg *et al.* [34] (figure 2*a*). We varied three model parameters during optimization ($F_{max}$, $\epsilon_{ref}$ and $K_{ff}$).

The parameters $F_{\max}$ and $\epsilon_{\mathrm{ref}}$ are analogous to maximum muscle isometric force and a tendon strain multiplier, respectively, with $K_{\mathrm{ff}}$ acting as a feed-forward gain. Parametrizing the controller in such a way allowed for large variability in the resulting behaviour of the ankle-foot prosthesis.

### 4.6.3. Balance controller

In this control architecture, prosthesis inversion or eversion torque was held constant during each stance phase and was calculated by adding a nominal torque parameter $\tau_{\mathrm{nom}}$ to a correctional term determined by multiplying a gain $K$ by centre of mass velocity deviations. To implement this controller, we made two modifications to the balance controller previously described in Kim *et al.* [25] (figure 3*a*). First, we measured deviations of centre of mass velocity instead of centre of mass acceleration. Centre of mass velocity was determined using two string potentiometers attached to the participant's waist and grounded to the treadmill handlebars. Since the side-to-side distance of the treadmill is known, we could calculate the centre of mass of the participant with additional information from the potentiometers. By differentiating the centre of mass position, we found the centre of mass velocity. Second, prior work relied on an instrumented treadmill to calculate the centre of mass deviation at the moment of intact limb toe-off, used to determine the ankle torque to be applied during the subsequent stance phase of the prosthetic foot. In this case study, we computed the centre of mass deviation when the prosthesis achieved foot flat due to a lack of an instrumented treadmill.

### 4.6.4. Time-based torque controller

Five parameters defined the torque profile of the time-based torque controller: device stiffness, peak torque magnitude, time of peak torque, rise time to peak torque and fall time after peak torque. Peak time, rise time, fall time and peak torque parameters are identical to previous work with exoskeletons [30]. For our work with the ankle-foot prosthesis, we also added an underlying baseline stiffness of the device (P1), creating a total of five optimization parameters (figure 4*a*). Although all parameters could be modified, three case studies utilized a controller where the peak torque was fixed at $0.5\ \mathrm{N\,m\,kg^{-1}}$, and only the other four parameters were optimized.

## 4.7. Preference optimization protocol

This case study focused on modifying the parameters of the four-parameter time-based torque controller in order to maximize subject comfort, as opposed to minimizing metabolic cost. The optimization protocol included six generations with five control laws evaluated per generation. However, instead of experiencing each control law only once, the participant experienced each control law multiple times for 23 s each. Starting with the presentation of the second control law, the participant was asked to rate the current control law as better or worse than the previous one. Control laws were presented in an order such that all five were directly compared with each other (12345135241), and a total preference score for each control law was determined at the end of the generation. This was then used to determine the distribution of the parameter set for the next generation. Three controllers were evaluated in validation: the optimized controller, a generic controller, and a controller with the parameters used for the optimization seed. To allow for comparison of all three controllers with one another, validation consisted of 5 min of continuous walking, during which time the participant was allowed to request any of the three controllers at any time. At the end of the 5 min trial, the participant ranked all three in order of preference.

Ethics. All study protocols were approved by the Institutional Review Board of either Carnegie Mellon University (Protocol ID IRBSTUDY2015_00000235) or Stanford University (Protocol ID 44162), depending on the location at which the study took place.

Data accessibility. The data and analysis scripts associated with this article are available as electronic supplementary material.

Authors' contributions. S.H.C., V.L.C. and A.S.V. conceived the experiments, V.L.C. and A.S.V. conducted the experiments, C.G.W., A.S.V. and V.L.C. analysed the results. C.G.W. wrote the manuscript. All authors edited and reviewed the manuscript.

Competing interests. We declare we have no competing interests.

Funding. This work was funded by a National Science Foundation Graduate Research Fellowship to C.G.W. (DGE-1656518) and National Science Foundation grant nos. CBET-1511177 and CMMI-1734449.

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
