## [Peer Review File · Royal Society Open Science]

Review History

RSOS-202020.R0 (Original submission)

Review form: Reviewer 1 (Max Shepherd)

Is the manuscript scientifically sound in its present form?

Yes

Are the interpretations and conclusions justified by the results?

Yes

Is the language acceptable?

Yes

Do you have any ethical concerns with this paper?

No

Have you any concerns about statistical analyses in this paper?

No

Recommendation?

Accept with minor revision (please list in comments)

Comments to the Author(s)

This paper describes several case studies in which the authors unsuccessfully attempted to improve the metabolic cost of walking with various controllers and control parameterizations. I think the negative results will be valuable for the field, and applaud them for sharing these attempts. I have minor comments/suggestions, detailed below.

Overall

The use of “damping”/“damper” in the Heel Stiffness Controller is inaccurate, as it’s not behaving like an actual damper. I recommend finding a different phrasing for this term... (work constant?)

The ordering of figures and tables make it a bit difficult to follow. I recommend moving Tables 1 and 2 closer to where the controllers are referenced in the text and removing the first reference to Fig 5 (page 3 line 40)

I recommend making sure that a reader can understand most of what’s needed for each control law without having to jump to the methods. Due to the nature of this paper, I think it’s okay for the Results section to be a bit heavier than typical with methods like controller descriptions. This would mostly require explaining the keel’s control law in the Heel Stiffness Controller and Balance Controller, and describing some of the parameters more thoroughly.

Introduction:

Page 2, Line 22: “Lack of pushoff work of passive prosthesis has been shown...” This phrasing it a little awkward – consider revising

Page 2 line 28: Citation #19 is not supporting the statement, and does not address energy expenditure

Page 2 line 28: While citation #18 does show metabolic improvements of 12% as the highest speed, it might be more appropriate to list the average findings, given the study was relatively low power. I recommend referring to these and similar results as generally mixed, and including this citation which found no metabolic benefit, with a slightly larger subject pool: “A controlled clinical trial of a clinically-tuned powered ankle prosthesis in people with transtibial amputation” Clinical rehabilitation, 2018

A recent high-quality study showed the metabolic benefits of passive damping in series: “Energy cost of ambulation in trans-tibial amputees using a dynamic-response foot with hydraulic versus rigid 'ankle': Insights from body centre of mass dynamics” JNER 2019

Page 3, Line 31-34: I recommend either removing this last point about rather dramatic simulation results or substantially bolstering it with further citations – it stands out from the previous results-driven discussion, which I believe does a good job already of illustrating that there is some promise, while implicitly acknowledging the difficulty of showing real positive results.

Page 3, line 55-57: “However, there was one exception where the participant completed an extended protocol to determine the effects of increased training on adaptation and optimization” –I think it would be helpful to go ahead and mention which protocol that was here.

I recommend touching on the large base of work that has sought (and largely failed) to find metabolic differences between passive feet. While the goal of this introduction is and should be to frame metabolic improvements as plausible and beneficial, particularly for active feet, it would help readers to understand that metabolic cost in amputees appears to be largely insensitive to passive mechanics.

Results:

Fig 1 caption: “damping from -0.5 to 0.5” –It’s unclear how to interpret this, without providing a little more context (maybe use different colors to show damping of, for instance 0.5 and -0.3?)

For the parameterizations with only 2 DOF, consider showing a visualization of metabolic cost vs the two parameters as a separate panel in the figure, to provide readers some intuition about the steepness vs noise of the landscape.

Page 4, line 36: Citation 27 – This is a rather old and not particularly often-cited article that shows higher hysteresis than most other ESR foot characterization papers. See “Instantaneous stiffness and hysteresis of dynamic elastic response prosthetic feet” *Prosthetics and Orthotics International*, 2017. Further, the amount of hysteresis varies quite a bit between feet, and will be affected by foot shell and shoe. I suggest removing this comparison with a “generic controller,” as it is not well motivated. This admittedly makes the overall story comparing generic controllers to optimized controllers harder to tell, but the motivation for this generic controller seems particularly weak in this instance.

Page 4, Line 54: The “K value” and “t_nom” should be briefly introduced conceptually. A zero-gain K seems quite different from a typical prosthesis, if I am reading this correctly. Does that mean the ankle has free inversion/eversion and cannot support a torque (other than the controlled bias torque?) To me this seems less like a “generic” version than ~infinite stiffness. The Antero-posterior control/mechanics (passive springs) for the balance controller should also be mentioned in this controller’s introduction in the results section (and in more detail in the methods if necessary), for readers to understand the dominant mechanics.

Page 5, line 55+, 4-parameter section: Please briefly mention or cite where the generic control law came from

Fig 4B: I found it difficult to tell which pane is associated with which subject/experiment. Please revise for clarity.

Fig 4C: What are the different groups of bar graphs? Are they supposed to have the same grouping as 4B columns? Consider clarifying.

Discussion

Page 10, lines 24-29. “In contrast, proprioception in people with amputation is limited to the residual limb and the interactions at the socket.” This might be oversimplified a bit, as upstream kinematics in the residual limb knee and hip, as well as whole-body proprioception may play a role in amputees’ ability to sense prosthesis mechanics. Amputees can rather precisely sense the mechanics of their prosthesis (see, for instance, “Amputee perception of prosthetic ankle stiffness during locomotion,” *JNER* 2018.)

Page 10, line 25: “It is possible that additional sensory feedback or direct control would improve adaptation and gait modification, and in turn reduce metabolic cost.” You may consider citing “Sensory feedback restoration in leg amputees improves walking speed, metabolic cost and phantom pain”, *Nature Medicine*, 2019

Page 10, line 33-34, “Specifically, past studies that showed successful reductions in metabolic cost in response to an active ankle-foot prosthesis have relied on a long adaptation protocol” Consider citing “Prosthetic energy return during walking increases after 3 weeks of adaptation to a new device” *JNER* 2018

Methods:

Page 13, line 6, “while stiffness dictates the torque provided at $0.7 \text{ N} \cdot \text{m}$ ” (This should be radians)

Review form: Reviewer 2

Is the manuscript scientifically sound in its present form?

Yes

Are the interpretations and conclusions justified by the results?

Yes

Is the language acceptable?

Yes

Do you have any ethical concerns with this paper?

No

Have you any concerns about statistical analyses in this paper?

No

Recommendation?

Major revision is needed (please make suggestions in comments)

Comments to the Author(s)

This manuscript describes a series of case studies attempting to optimize control of an active prosthetic foot emulator to minimize energy cost of walking in people with a transtibial amputation. I appreciate the well-structured description of the application and experiences in this pilot-like experiment. Although case studies do not offer the highest level of evidence in clinical research, they can provide valuable insights and information. We tend to miss many of these efforts and experiences in literature leading to potential a publication bias.

Nevertheless, there are several issues that should be addressed to ensure impact of this manuscript.

Major concerns

- It seems that the number of generations and number of iterations with each generation were decided and fixed at the start of each case. It can be wondered whether saturation of the optimal solution found at the end of the protocol occurred. No information is provided on the development of energy cost over the duration of the protocol, which makes it impossible to assess this. In the discussion authors seem to indicate that the number of generations and iterations with generations selected resulted in saturation in a previous study, but this not guarantee that saturation occurred in these cases. In fact, in several cases the optimized control resulted in higher energy cost than the generic control, showing that control was not optimal.
- Generic control was used as a reference condition, next to the prescribed prosthesis, it seems that this generic control was quite arbitrarily selected for each case. Why did you not choose the initial condition control as reference. In that way it could have been seen how much energy cost changed (decreased) at the end of the optimization process.
- When comparing the results of the emulator (with optimized of generic control) with the prescribed foot. Did you correct for differences in mass of the prosthesis. And could additional differences between the emulator and prescribed prosthesis account for the higher energy cost with the emulator? For instance: roll over profile or alignment of the foot relative to the socket?
- Energy cost is used as the cost function in optimization (and evaluation) . The authors refer to some limitation in this cost function. Limited reliability might be one (both limited reproducibility between trials as well as the error introduced in curve fitting beyond 2 min). Could the authors envisage the use of other cost functions in the future, potentially related to gait speed or stability?

Minor comments

- page 5 line 31 : could the heel stiffness control condition also result in net positive work generation (negative damping) . Could you explain why this was not found in the optimized condition?
- Page 6, line 37: Am I correct to conclude that the optimal solution in the 5 parameter controller preferred the behaviour of a passive ankle spring, as peak active torque was negligible? Could you explain that finding?

- page 6, line 55: why did you not use similar generic control for reference in the 4 and 5 parameter optimization?
- page 7, line 50: on what did you base the parameters of the generic control condition in this case?
- page 9, line 23: what do you mean by 'optimization seed'. Is that the initial condition of the optimization process
- page 9, line 35: avoid the word "significant change" here as it usually refers to statistical testing, which was not performed in this study. Use something like "consistent improvements"
- page 10 line 28: Better convergence to an optimal solution does not only depend on the time of each condition (as in extended protocol) but also on the number of generations and number of iterations per generation I believe.
- page 10, line 35: what do you mean by "metabolic cost variation"
- Page 11, paragraph 2: I'm not convinced that the lack of the ankle (compared to exoskeleton users) provides less sensory feedback. It is unclear whether and how torque enhancement at the ankle is sensed by ankle proprioceptors or musclesensors. It might well be sensed in the interface between device and leg as in the socket of the prosthesis. Moreover, control might be easier when the device does not have to cooperate with the biological ankle muscle but has a one-to-one effect to ankle torque provided?
- page 11, line 42: prioritizing stability might also require extra metabolic energy. Hence better stability might provide lower metabolic cost?
- page 11, line 44: "There are a myriad of reasons why HILO may be more challenging when applied to powered prostheses instead of exoskeletons" maybe the biggest difference is not in the device but in the fact that these exoskeleton studies are generally performed in able-bodied persons?
- Page 13, line 15: visualize or describe this optimization process, outline what is meant by generation, control law and provide an example of how energy cost converged to an optimum in this process.
- Page 13, line 45-48: define heel angle and describe how inversion-eversion is controlled in the 3DOF foot.
- Page 14, line 43: explain the parameters in the balance controller (K and tau) in little more detail.

Decision letter (RSOS-202020.R0)

Dear Ms Welker

The Editors assigned to your paper RSOS-202020 "Shortcomings of human-in-the-loop optimization for an ankle-foot prosthesis emulator: a case series" have now received comments from reviewers and would like you to revise the paper in accordance with the reviewer comments and any comments from the Editors. Please note this decision does not guarantee eventual acceptance.

Please submit your revised manuscript and required files (see below) no later than 21 days from today's (ie 03-Feb-2021) date. Note: the ScholarOne system will 'lock' if submission of the revision is attempted 21 or more days after the deadline. If you do not think you will be able to meet this deadline please contact the editorial office immediately.

on behalf of Prof Kevin Padian (Subject Editor)
openscience@royalsociety.org

Associate Editor Comments to Author:

Thank you for the submitting this paper to the journal. The reviewers think the work has merit, and the journal team are glad to see a group trying to publish negative outcomes in the journal - we encourage this, but they remain rare (no doubt for established publishing and research assessment cultures that discourage them over new, exciting results). In any case, we second the reviewer comment that this is to be applauded, and encourage you to engage closely with the reviewer comments in your revision - we do not generally permit multiple rounds of revision, so take time to carefully address the concerns. Good luck and thanks again.

Editor comments

Thanks for your submission and best wishes for your revisions. If you need more time please contact the editorial office. We ask you to address the comments of the reviewers thoroughly.

Reviewer comments to Author:

Reviewer: 1

Comments to the Author(s)

This paper describes several case studies in which the authors unsuccessfully attempted to improve the metabolic cost of walking with various controllers and control parameterizations. I think the negative results will be valuable for the field, and applaud them for sharing these attempts. I have minor comments/suggestions, detailed below.

Overall

The use of "damping"/"damper" in the Heel Stiffness Controller is inaccurate, as it's not behaving like an actual damper. I recommend finding a different phrasing for this term... (work constant?)

The ordering of figures and tables make it a bit difficult to follow. I recommend moving Tables 1 and 2 closer to where the controllers are referenced in the text and removing the first reference to Fig 5 (page 3 line 40)

I recommend making sure that a reader can understand most of what's needed for each control law without having to jump to the methods. Due to the nature of this paper, I think it's okay for the Results section to be a bit heavier than typical with methods like controller descriptions. This would mostly require explaining the keel's control law in the Heel Stiffness Controller and Balance Controller, and describing some of the parameters more thoroughly.

Introduction:

Page 2, Line 22: "Lack of pushoff work of passive prosthesis has been shown..." This phrasing is a little awkward – consider revising

Page 2 line 28: Citation #19 is not supporting the statement, and does not address energy expenditure

Page 2 line 28: While citation #18 does show metabolic improvements of 12% as the highest speed, it might be more appropriate to list the average findings, given the study was relatively low power. I recommend referring to these and similar results as generally mixed, and including this citation which found no metabolic benefit, with a slightly larger subject pool: "A controlled clinical trial of a clinically-tuned powered ankle prosthesis in people with transtibial amputation" Clinical rehabilitation, 2018

A recent high-quality study showed the metabolic benefits of passive damping in series: "Energy cost of ambulation in trans-tibial amputees using a dynamic-response foot with hydraulic versus rigid 'ankle': Insights from body centre of mass dynamics" JNER 2019

Page 3, Line 31-34: I recommend either removing this last point about rather dramatic simulation results or substantially bolstering it with further citations – it stands out from the previous results-driven discussion, which I believe does a good job already of illustrating that there is some promise, while implicitly acknowledging the difficulty of showing real positive results.

Page 3, line 55-57: "However, there was one exception where the participant completed an extended protocol to determine the effects of increased training on adaptation and optimization" –I think it would be helpful to go ahead and mention which protocol that was here.

I recommend touching on the large base of work that has sought (and largely failed) to find metabolic differences between passive feet. While the goal of this introduction is and should be to frame metabolic improvements as plausible and beneficial, particularly for active feet, it would help readers to understand that metabolic cost in amputees appears to be largely insensitive to passive mechanics.

Results:

Fig 1 caption: "damping from -0.5 to 0.5" –It's unclear how to interpret this, without providing a little more context (maybe use different colors to show damping of, for instance 0.5 and -0.3?)

For the parameterizations with only 2 DOF, consider showing a visualization of metabolic cost vs the two parameters as a separate panel in the figure, to provide readers some intuition about the steepness vs noise of the landscape.

Page 4, line 36: Citation 27 – This is a rather old and not particularly often-cited article that shows higher hysteresis than most other ESR foot characterization papers. See "Instantaneous stiffness and hysteresis of dynamic elastic response prosthetic feet" Prosthetics and Orthotics International, 2017. Further, the amount of hysteresis varies quite a bit between feet, and will be affected by foot shell and shoe. I suggest removing this comparison with a "generic controller,"

as it is not well motivated. This admittedly makes the overall story comparing generic controllers to optimized controllers harder to tell, but the motivation for this generic controller seems particularly weak in this instance.

Page 4, Line 54: The "K value" and "t_nom" should be briefly introduced conceptually.

A zero-gain K seems quite different from a typical prosthesis, if I am reading this correctly. Does that mean the ankle has free inversion/eversion and cannot support a torque (other than the controlled bias torque?) To me this seems less like a “generic” version than ~infinite stiffness. The Antero-posterior control/mechanics (passive springs) for the balance controller should also be mentioned in this controller’s introduction in the results section (and in more detail in the methods if necessary), for readers to understand the dominant mechanics.

Page 5, line 55+, 4-parameter section: Please briefly mention or cite where the generic control law came from

Fig 4B: I found it difficult to tell which pane is associated with which subject/experiment. Please revise for clarity.

Fig 4C: What are the different groups of bar graphs? Are they supposed to have the same grouping as 4B columns? Consider clarifying.

Discussion

Page 10, lines 24-29. “In contrast, proprioception in people with amputation is limited to the residual limb and the interactions at the socket.” This might be oversimplified a bit, as upstream kinematics in the residual limb knee and hip, as well as whole-body proprioception may play a role in amputees’ ability to sense prosthesis mechanics. Amputees can rather precisely sense the mechanics of their prosthesis (see, for instance, “Amputee perception of prosthetic ankle stiffness during locomotion,” JNER 2018.)

Page 10, line 25: “It is possible that additional sensory feedback or direct control would improve adaptation and gait modification, and in turn reduce metabolic cost.” You may consider citing “Sensory feedback restoration in leg amputees improves walking speed, metabolic cost and phantom pain”, Nature Medicine, 2019

Page 10, line 33-34, “Specifically, past studies that showed successful reductions in metabolic cost in response to an active ankle-foot prosthesis have relied on a long adaptation protocol” Consider citing “Prosthetic energy return during walking increases after 3 weeks of adaptation to a new device” JNER 2018

Methods:

Page 13, line 6, “while stiffness dictates the torque provided at $0.7 \text{ N} \cdot \text{m}$ ” (This should be radians)

Reviewer: 2

Comments to the Author(s)

This manuscript describes a series of case studies attempting to optimize control of an active prosthetic foot emulator to minimize energy cost of walking in people with a transtibial amputation. I appreciate the well-structured description of the application and experiences in this pilot-like experiment. Although case studies do not offer the highest level of evidence in clinical research, they can provide valuable insights and information. We tend to miss many of these efforts and experiences in literature leading to potential a publication bias.

Nevertheless, there are several issues that should be addressed to ensure impact of this manuscript.

Major concerns

- It seems that the number of generations and number of iterations with each generation were decided and fixed at the start of each case. It can be wondered whether saturation of the optimal solution found at the end of the protocol occurred. No information is provided on the development of energy cost over the duration of the protocol, which makes it impossible to assess this. In the discussion authors seem to indicate that the number of generations and iterations with generations selected resulted in saturation in a previous study, but this not guarantee that

saturation occurred in these cases. In fact, in several cases the optimized control resulted in higher energy cost than the generic control, showing that control was not optimal.

- Generic control was used as a reference condition, next to the prescribed prosthesis, it seems that this generic control was quite arbitrarily selected for each case. Why did you not choose the initial condition control as reference. In that way it could have been seen how much energy cost changed (decreased) at the end of the optimization process.
 - When comparing the results of the emulator (with optimized of generic control) with the prescribed foot. Did you correct for differences in mass of the prosthesis. And could additional differences between the emulator and prescribed prosthesis account for the higher energy cost with the emulator? For instance: roll over profile or alignment of the foot relative to the socket?
 - Energy cost is used as the cost function in optimization (and evaluation) . The authors refer to some limitation in this cost function. Limited reliability might be one (both limited reproducibility between trials as well as the error introduced in curve fitting beyond 2 min).
- Could the authors envisage the use of other cost functions in the future, potentially related to gait speed or stability?

Minor comments

- page 5 line 31 : could the heel stiffness control condition also result in net positive work generation (negative damping) . Could you explain why this was not found in the optimized condition?
- Page 6, line 37: Am I correct to conclude that the optimal solution in the 5 parameter controller preferred the behaviour of a passive ankle spring, as peak active torque was negligible? Could you explain that finding?
- page 6, line 55: why did you not use similar generic control for reference in the 4 and 5 parameter optimization?
- page 7, line 50: on what did you base the parameters of the generic control condition in this case?
- page 9, line 23: what do you mean by 'optimization seed'. Is that the initial condition of the optimization process
- page 9, line 35: avoid the word "significant change" here as it usually refers to statistical testing, which was not performed in this study. Use something like "consistent improvements"
- page 10 line 28: Better convergence to an optimal solution does not only depend on the time of each condition (as in extended protocol) but also on the number of generations and number of iterations per generation I believe.
- page 10, line 35: what do you mean by "metabolic cost variation"
- Page 11, paragraph 2: I'm not convinced that the lack of the ankle (compared to exoskeleton users) provides less sensory feedback. It is unclear whether and how torque enhancement at the ankle is sensed by ankle proprioceptors or musclesensors. It might well be sensed in the interface between device and leg as in the socket of the prosthesis. Moreover, control might be easier when the device does not have to cooperate with the biological ankle muscle but has a one-to-one effect to ankle torque provided?
- page 11, line 42: prioritizing stability might also require extra metabolic energy. Hence better stability might provide lower metabolic cost?
- page 11, line 44: "There are a myriad of reasons why HILO may be more challenging when applied to powered prostheses instead of exoskeletons" maybe the biggest difference is not in the device but in the fact that these exoskeleton studies are generally performed in able-bodied persons?
- Page 13, line 15: visualize or describe this optimization process, outline what is meant by generation, control law and provide an example of how energy cost converged to an optimum in this process.
- Page 13, line 45-48: define heel angle and describe how inversion-eversion is controlled in the 3DOF foot.

- Page 14, line 43: explain the parameters in the balance controller (K and τ) in little more detail.

===PREPARING YOUR MANUSCRIPT===

===PREPARING YOUR REVISION IN SCHOLARONE===

- 1) One version identifying all the changes that have been made (for instance, in coloured highlight, in bold text, or tracked changes);
 - 2) A 'clean' version of the new manuscript that incorporates the changes made, but does not highlight them.
 - An individual file of each figure (EPS or print-quality PDF preferred [either format should be produced directly from original creation package], or original software format).
 - An editable file of each table (.doc, .docx, .xls, .xlsx, or .csv).
 - An editable file of all figure and table captions.
- Note: you may upload the figure, table, and caption files in a single Zip folder.
- Any electronic supplementary material (ESM).
 - If you are requesting a discretionary waiver for the article processing charge, the waiver form must be included at this step.
 - If you are providing image files for potential cover images, please upload these at this step, and inform the editorial office you have done so. You must hold the copyright to any image provided.
 - A copy of your point-by-point response to referees and Editors. This will expedite the preparation of your proof.

- Ensure that your data access statement meets the requirements at <https://royalsociety.org/journals/authors/author-guidelines/#data>. You should ensure that you cite the dataset in your reference list. If you have deposited data etc in the Dryad repository, please include both the 'For publication' link and 'For review' link at this stage.
- If you are requesting an article processing charge waiver, you must select the relevant waiver option (if requesting a discretionary waiver, the form should have been uploaded at Step 3 'File upload' above).
- If you have uploaded ESM files, please ensure you follow the guidance at <https://royalsociety.org/journals/authors/author-guidelines/#supplementary-material> to include a suitable title and informative caption. An example of appropriate titling and captioning may be found at https://figshare.com/articles/Table_S2_from_Is_there_a_trade-off_between_peak_performance_and_performance_breadth_across_temperatures_for_aerobic_sc_ope_in_teleost_fishes_/3843624.

Author's Response to Decision Letter for (RSOS-202020.R0)

See Appendix A.

RSOS-202020.R1 (Revision)

Review form: Reviewer 2

Is the manuscript scientifically sound in its present form?

Yes

Are the interpretations and conclusions justified by the results?

Yes

Is the language acceptable?

Yes

Do you have any ethical concerns with this paper?

No

Have you any concerns about statistical analyses in this paper?

No

Recommendation?

Accept as is

Comments to the Author(s)

Thanks you for your adequate reply to my questions and concerns. They clarified important issues and contribute to a very interesting and relevant manuscript. I have no further comments

Decision letter (RSOS-202020.R1)

Dear Ms Welker,

It is a pleasure to accept your manuscript entitled "Shortcomings of human-in-the-loop optimization for an ankle-foot prosthesis emulator: a case series" in its current form for publication in Royal Society Open Science. The comments of the reviewer(s) who reviewed your manuscript are included at the foot of this letter.

Please see the Royal Society Publishing guidance on how you may share your accepted author manuscript at <https://royalsociety.org/journals/ethics-policies/media-embargo/>. After publication, some additional ways to effectively promote your article can also be found here

<https://royalsociety.org/blog/2020/07/promoting-your-latest-paper-and-tracking-your-results/>.

on behalf of Kevin Padian (Subject Editor)
openscience@royalsociety.org

Associate Editor Comments to Author:

Comments to the Author:

The comments of the reviewer have been satisfactorily addressed and the paper may be accepted - congratulations!

Reviewer comments to Author:

Reviewer: 2

Comments to the Author(s)

Thanks you for your adequate reply to my questions and concerns. They clarified important issues and contribute to a very interesting and relevant manuscript. I have no further comments

Appendix A

Dear Royal Society Open Science Editorial Board and Reviewers,

Thank you for the thoughtful and constructive review of our paper, which has significantly improved our manuscript. Below you will find each reviewer comment, along with the response (in green text), and the accompanying changes to the manuscript (in blue text). In addition, the revised version of the manuscript highlights all new text in blue.

Associate Editor

Associate Editor Comments to Author:

Comment E1-1: Thank you for the submitting this paper to the journal. The reviewers think the work has merit, and the journal team are glad to see a group trying to publish negative outcomes in the journal - we encourage this, but they remain rare (no doubt for established publishing and research assessment cultures that discourage them over new, exciting results). In any case, we second the reviewer comment that this is to be applauded, and encourage you to engage closely with the reviewer comments in your revision - we do not generally permit multiple rounds of revision, so take time to carefully address the concerns. Good luck and thanks again.

- **Response:** Thank you; we hope that these negative results will be helpful to those in the field undertaking similar endeavors. We appreciate the deadline extension and now feel that we have been able to carefully address both reviewers' concerns.

Reviewer: 1

Comments to the Author(s)

Comment R1-1: This paper describes several case studies in which the authors unsuccessfully attempted to improve the metabolic cost of walking with various controllers and control parameterizations. I think the negative results will be valuable for the field, and applaud them for sharing these attempts. I have minor comments/suggestions, detailed below.

- **Response:** Thank you for your helpful suggestions; we are confident that the manuscript has improved as a result of the changes made.

Overall

Comment R1-2: The use of “damping”/”damper” in the Heel Stiffness Controller is inaccurate, as it’s not behaving like an actual damper. I recommend finding a different phrasing for this term... (work constant?)

- **Response:** We agree that this term should be changed to avoid confusion. We have replaced all references of the damping coefficient with a work constant coefficient instead.

Comment R1-3: The ordering of figures and tables make it a bit difficult to follow. I recommend moving Tables 1 and 2 closer to where the controllers are referenced in the text and removing the first reference to Fig 5 (page 3 line 40)

- Response: Thank you for pointing this out. With this comment in mind, we have moved the table with participant information to the methods section at the end of the paper so that the table presenting an overview of the case studies completed is closer to where the controller results presented in the text. In addition, we have removed the first reference to Fig 5, as suggested.

Comment R1-4: I recommend making sure that a reader can understand most of what's needed for each control law without having to jump to the methods. Due to the nature of this paper, I think it's okay for the Results section to be a bit heavier than typical with methods like controller descriptions. This would mostly require explaining the keel's control law in the Heel Stiffness Controller and Balance Controller, and describing some of the parameters more thoroughly.

- Response: We agree that requiring the reader to repeatedly refer to the methods at the end of the paper was undesirable. We have added an additional sentence or two at the beginning of each controller results section providing further details regarding the parameters optimized in the controller, as follows:

Heel Stiffness Controller

Two parameters dictating the torque trajectory of the prosthesis heel were optimized in the case study of this novel controller: a heel stiffness parameter and a heel work constant that dictated the amount of work provided or dissipated by the heel during the gait cycle over the course of loading and unloading (Fig. 1A)

Neuromuscular Controller

Three parameters in the neuromuscular controller based on previous literature [33] were optimized. Two were parameters within the muscle-tendon complex model: (1) F_{max} , which is analogous to maximum muscle isometric force, and (2) E_{ref} , which acts as a tendon strain multiplier. The third parameter was a feedforward gain K_{ff} , which affected the magnitude of the muscle activation (Fig. 2A).

Balance Controller

In this control architecture, the prosthesis behaved as a passive spring in plantarflexion and dorsiflexion, while a baseline nominal inversion/eversion torque, t_{nom} , and a gain dictating the magnitude of correction for center of mass velocity deviations, K , were optimized. (Fig. 3A).

Time-based Torque Controller

The parameterization of this controller was based on a similar controller successfully used to optimize the parameters of ankle exoskeleton torque to reduce metabolic cost [30], with an additional underlying baseline prosthesis stiffness parameter... The five parameters optimized in this control architecture were: baseline stiffness, and the peak time, rise time, fall time, and peak magnitude of the additive torque (Fig. 4A).

Introduction:

Comment R1-5: Page 2, Line 22: “Lack of pushoff work of passive prosthesis has been shown...” This phrasing is a little awkward—consider revising

- Response: We have revised the sentence as follows:

Lack of push-off work in passive prostheses is implicated in higher step-to-step transition losses [18-19], which are correlated with increased metabolic rate [20].

Comment R1-6: Page 2 line 28: Citation #19 is not supporting the statement, and does not address energy expenditure

- Response: Thank you for pointing out this error. We no longer cite this reference to support this statement.

Comment R1-7: Page 2 line 28: While citation #18 does show metabolic improvements of 12% as the highest speed, it might be more appropriate to list the average findings, given the study was relatively low power. I recommend referring to these and similar results as generally mixed, and including this citation which found no metabolic benefit, with a slightly larger subject pool: “A controlled clinical trial of a clinically-tuned powered ankle prosthesis in people with transtibial amputation” Clinical rehabilitation, 2018

- Response: We agree and have amended the sentence by citing the more modest average reductions and adding the suggested citation as follows:

A device providing push-off work and emulating biological characteristics of human neuromuscular control has resulted in mixed results with respect to metabolic cost, with some studies finding modest reductions [e.g. 8% in 23] and other studies finding no significant differences compared to walking with a passive prosthesis [24].

Comment R1-8: A recent high-quality study showed the metabolic benefits of passive damping in series: “Energy cost of ambulation in trans-tibial amputees using a dynamic-response foot with hydraulic versus rigid 'ankle': Insights from body centre of mass dynamics” JNER 2019

- Response: Thank you for pointing out this article. We have added this citation in order to motivate the heel stiffness controller, which incorporates an ankle mechanism comparable to that of this type of passive prosthesis:

Four different classes of control architecture were tested: (1) a heel stiffness controller that varied the stiffness and damping of the heel of the prosthesis (Fig. 1A), inspired by the observation that damped articulation of the ankle can reduce energy cost compared to rigid prosthetic ankles [33]...

Comment R1-9: Page 3, Line 31-34: I recommend either removing this last point about rather dramatic simulation results or substantially bolstering it with further citations—it stands out from the previous results-driven discussion, which I believe does a good job already of illustrating that there is some promise, while implicitly acknowledging the difficulty of showing real positive results.

- Response: Thank you for your comment. We think that it is important to discuss the theoretical upper limit of benefits that may be obtained by passive prosthetic devices. We have added a citation discussing the role of ankle power in human walking, and have revised our discussion of the simulation results as follows:

The experimental results of active prostheses show some promise, but biomechanical analyses and simulations suggest that active prostheses have more potential to mitigate problems faced by individuals with amputation. The ankle is estimated to provide approximately half of the power needed for healthy human walking [26], much of which is not replaced by passive prostheses. Simulation models suggest that active prostheses have the potential to reduce the metabolic cost of walking significantly below that of unimpaired walking [27].

[26] Ferris et al. 2012. Evaluation of a Powered Ankle-Foot Prosthetic System During Walking. *Archives of Physical Medicine and Rehabilitation*. 93, 1911-1918.

Comment R1-10: Page 3, line 55-57: “However, there was one exception where the participant completed an extended protocol to determine the effects of increased training on adaptation and optimization” –I think it would be helpful to go ahead and mention which protocol that was here.

- Response: We agree that this is an important point to clarify. We have revised the relevant sentence to include a comparison of the time spent in each condition for the extended protocol compared to the typical protocol as follows:

However, there was one exception where the participant completed an extended protocol, in which the time spent in each condition was increased by a factor of five, to determine the effects of increased training on adaptation and optimization.

Comment R1-11: I recommend touching on the large base of work that has sought (and largely failed) to find metabolic differences between passive feet. While the goal of this introduction is and should be to frame metabolic improvements as plausible and beneficial, particularly for active feet, it would help readers to understand that metabolic cost in amputees appears to be largely insensitive to passive mechanics.

- Response: We agree that this is an important point to mention in the introduction. Because some studies have found metabolic differences when varying passive prosthesis parameters such as stiffness, we add a sentence discussing the mixed results from these studies as follows:

Previous work has found that tuning passive prosthesis parameters can have a small effect on metabolic cost [8-9], while other studies have found metabolic cost to be unaffected [10-11].

[8] Zelik et al. 2011. Systematic variation of prosthetic foot spring affects center-of-mass mechanics and metabolic cost during walking. *IEEE Transactions on Neural Systems and Rehabilitation Engineering* **19**, 411-419.

[9] Major et al. 2014. The effects of prosthetic ankle stiffness on ankle and knee kinematics, prosthetic limb loading, and net metabolic cost of trans-tibial amputee gait. *Clinical Biomechanics* **29**, 98-104.

[10] Gailey et al. 1997. The effects of prosthesis mass on metabolic cost of ambulation in non-vascular trans-tibial amputees. *Prosthetics and Orthotics International* **21**, 9-16.

[11] Beck et al. 2017. Prosthetic model, but not stiffness or height, affects the metabolic cost of running for athletes with transtibial amputations. *Journal of Applied Physiology* **123**.

Results:

Comment R1-12: Fig 1 caption: “damping from -0.5 to 0.5” –It’s unclear how to interpret this, without providing a little more context (maybe use different colors to show damping of, for instance 0.5 and -0.3?)

- Response: We agree that this should be clarified and have revised the caption to provide more context as follows:

Stiffness was varied between 50 and 130 Nm/rad, and the work constant varied from -0.5 to 0.5, with a larger positive work constant resulting in more positive work injected during the gait cycle.

Comment R1-13: For the parameterizations with only 2 DOF, consider showing a visualization of metabolic cost vs the two parameters as a separate panel in the figure, to provide readers some intuition about the steepness vs noise of the landscape.

- Response: We agree that it would be helpful for the reader to get a sense of the magnitude of the noise relative to changes in metabolic cost due to changing prosthesis control parameters. In response to this Comment, as well as Comment R2-2, we now include supplementary figures visualizing the metabolic cost over time during optimization for two representative case studies to demonstrate that there was not a strong negative trend in metabolic rate over the course of the optimization. We prefer not to present metabolic rate versus parameters with landscape representation because we have previously found that this relationship is typically not constant, due to adaptation that occurs over time during the optimization process (Zhang et al, 2017).

Comment R1-14: Page 4, line 36: Citation 27—This is a rather old and not particularly often-cited article that shows higher hysteresis than most other ESR foot characterization papers. See “Instantaneous stiffness and hysteresis of dynamic elastic response prosthetic feet” *Prosthetics and Orthotics International*, 2017. Further, the amount of hysteresis varies quite a bit between feet, and will be affected by foot shell and shoe. I suggest removing this comparison with a “generic controller,” as it is not well motivated. This admittedly makes the overall story comparing generic controllers to optimized controllers harder to tell, but the motivation for this generic controller seems particularly weak in this instance.

- Response: Thank you for pointing out this important point to clarify. Although we motivated the chosen generic controllers from literature, we do not mean to claim that they are wholly representative of a given class of devices. Instead, we use the generic

control simply as a comparison point, as we had hypothesized that optimized control parameters would reduce metabolic cost compared to any other control parameters. We have revised this sentence to clarify this point as follows:

In validation, the optimized controller was compared to a generic controller (stiffness = 120 Nm, work constant = -0.5) chosen to provide 32% heel energy dissipation, which approximates some energy storage and return (ESR) devices with higher energy dissipation [35].

Comment R1-15: Page 4, Line 54: The “K value” and “t_nom” should be briefly introduced conceptually.

- Response: Thank you for pointing out that this should be clarified. We have modified this sentence to provide a brief introduction to these values as follows:

In this control architecture, the prosthesis behaved as a passive spring in plantarflexion and dorsiflexion, while a baseline nominal inversion/eversion torque, t_{nom} , and a gain dictating the magnitude of correction for center of mass velocity deviations, K , were optimized. (Fig. 3A).

Comment R1-16: A zero-gain K seems quite different from a typical prosthesis, if I am reading this correctly. Does that mean the ankle has free inversion/eversion and cannot support a torque (other than the controlled bias torque?) To me this seems less like a “generic” version than \sim infinite stiffness.

- Response: We apologize for the confusion. A zero-gain K in this instance means that the ankle is not providing corrections for deviations in center-of-mass velocity, but simply behaves as a passive spring in plantarflexion and dorsiflexion. We have clarified this in the methods as follows:

In this control architecture, prosthesis inversion or eversion torque was held constant during each stance phase and was calculated by adding a nominal torque parameter t_{nom} to a correctional term determined by multiplying a gain K by center of mass velocity deviations.

Comment R1-17: The Antero-posterior control/mechanics (passive springs) for the balance controller should also be mentioned in this controller’s introduction in the results section (and in more detail in the methods if necessary), for readers to understand the dominant mechanics.

- Response: We agree that this is an important point to clarify and have addressed this in the introductory sentence to the balance controller in the results section:

In this control architecture, the prosthesis behaved as a passive spring in plantarflexion and dorsiflexion, while a baseline nominal inversion/eversion torque, t_{nom} , and a gain dictating the magnitude of correction for center of mass velocity deviations, K , were optimized. (Fig. 3A).

Comment R1-18: Page 5, line 55+, 4-parameter section: Please briefly mention or cite where the generic control law came from

- Response: Thank you for pointing out that this should be clarified. We have added an introductory section to the time-based torque controller section describing the motivation behind this controller, as well as how the generic control parameters were selected:

In choosing the parameters for the generic controller, the stiffness parameter was based on the approximate stiffness of the participant's prescribed prosthesis, while the remaining magnitude and timing parameters were based on average optimal parameters for ankle exoskeletons, with subject-specific modifications made for subject comfort.

Comment R1-19: Fig 4B: I found it difficult to tell which pane is associated with which subject/experiment. Please revise for clarity.

- Response: Thank you for pointing this out. We have made multiple revisions to the figure in the hopes of adding more clarity. These changes include: combining panels B, C, and D to make the grouping between the optimization and validation results more clear, including subject information for all case studies, and separating the metabolic validation results into separate graphs, as depicted below:

Comment R1-20: Fig 4C: What are the different groups of bar graphs? Are they supposed to have the same grouping as 4B columns? Consider clarifying.

- Response: The bar graphs were intended to have the same grouping as the 4B columns. We hope that this figure has been clarified by combining panels B, C, and D, as well as including separate y-axes for the bar graphs to better signify that the labels associated with the optimization results also apply to the bar graphs of metabolic cost.

Discussion

Comment R1-21: Page 10, lines 24-29. "In contrast, proprioception in people with amputation is limited to the residual limb and the interactions at the socket." This might be oversimplified a bit, as upstream kinematics in the residual limb knee and hip, as well as whole-body proprioception may play a role in amputees' ability to sense prosthesis mechanics. Amputees can rather precisely sense the mechanics of their prosthesis (see, for instance, "Amputee perception of prosthetic ankle stiffness during locomotion," JNER 2018.)

- Results: Thank you for pointing out this clarification. We have modified this statement to include the nuances that you stated, as well as the citation to clarify that people with amputation are able to identify prosthesis stiffness fairly accurately as follows:

In contrast, proprioception in people with amputation is altered, since sensory information is not available directly from the limb. Although people with amputation can fairly precisely sense the stiffness of their prosthesis, sensory information must come from interactions at the socket and whole-body proprioception of resulting knee and hip kinematics [40].

Comment R1-22: Page 10, line 25: "It is possible that additional sensory feedback or direct control would improve adaptation and gait modification, and in turn reduce metabolic cost." You may consider citing "Sensory feedback restoration in leg amputees improves walking speed, metabolic cost and phantom pain", Nature Medicine, 2019

- Response: We thank you for pointing out this paper, but we have some concerns about the strength of claims made based on the evidence presented. Since the idea in the current paper is purely speculative, we prefer not to provide an accompanying citation.

Comment R1-23: Page 10, line 33-34, "Specifically, past studies that showed successful reductions in metabolic cost in response to an active ankle-foot prosthesis have relied on a long adaptation protocol" Consider citing "Prosthetic energy return during walking increases after 3 weeks of adaptation to a new device" JNER 2018

- Response: Thank you for pointing out this relevant citation. We have added it to the manuscript to support this statement.

Methods:

Comment R1-24: Page 13, line 6, "while stiffness dictates the torque provided at 0.7 N · m" (This should be radians)

- Response: Thank you for pointing out this error; it has been corrected.

Reviewer: 2

Comments to the Author(s)

Comment R2-1: This manuscript describes a series of case studies attempting to optimize control of an active prosthetic foot emulator to minimize energy cost of walking in people with a transtibial amputation. I appreciate the well-structured description of the application and experiences in this pilot-like experiment. Although case studies do not offer the highest level of evidence in clinical research, they can provide valuable insights and information. We tend to miss many of these efforts and experiences in literature leading to potential a publication bias. Nevertheless, there are several issues that should be addressed to ensure impact of this manuscript.

- Response: Thank you for your thoughtful feedback. We hope this manuscript detailing our negative results will be useful to those attempting similar work in the future and that the changes made in response to your comments has improved the manuscript.

Major concerns

Comment R2-2: It seems that the number of generations and number of iterations with each generation were decided and fixed at the start of each case. It can be wondered whether saturation of the optimal solution found at the end of the protocol occurred. No information is provided on the development of energy cost over the duration of the protocol, which makes it impossible to assess this. In the discussion authors seem to indicate that the number of generations and iterations with generations selected resulted in saturation in a previous study, but this not guarantee that saturation occurred in these cases. In fact, in several cases the optimized control resulted in higher energy cost than the generic control, showing that control was not optimal.

- Response: We agree that the parameters chosen by the optimizer were in fact not optimal, as the generic control outperformed optimized control in multiple case studies. Our intention in performing these studies was to replicate a protocol that can successfully find optimal control parameters in exoskeletons and apply it to prostheses. If the optimization procedure was functioning as intended, we would expect to see a decrease in metabolic cost over time when comparing each generation; in our prior experiments with exoskeletons, published and unpublished, large (10-25%) improvements in energy cost are evident within the first four generations of optimization (e.g. Zhang et al., 2017). However, we did not find this trend, and therefore did not continue the optimization protocol. We agree that information regarding the energy cost over the duration of the protocol would be helpful to include so that readers can judge this for themselves. We have therefore included supplemental figures for two representative case studies visualizing how metabolic rate varied over time for each control condition and generation. These figures demonstrate that there is no consistent decreasing trend of metabolic rate between generations. We have also been careful not to claim that the optimizer had fully converged.

Comment R2-3: Generic control was used as a reference condition, next to the prescribed prosthesis, it seems that this generic control was quite arbitrarily selected for each case. Why did you not choose the initial condition control as reference. In that way it could have been seen how much energy cost changed (decreased) at the end of the optimization process.

- Response: The purpose of choosing generic control parameters was simply to have a comparison point to optimized control parameters. No matter what parameters we chose for generic control, the purpose of the optimization was to find the parameters that result in the lowest metabolic cost (or the highest preference) compared to all other parameters. Therefore, any other parameters, as long as we did not happen to choose values selected as optimized parameters, would have been able to serve as this comparison point. In choosing the generic parameters, we did want to select within a range that had previously been used with each type of control. Therefore, we based these generic control parameters on literature, with subject-specific modifications made. We have clarified our intentions in the comparison between the optimized and generic control parameters as follows:

Generic control parameters were based on those used in literature with similar controllers, with subject-specific modifications made in each case study. Based on previous success of human-in-the-loop optimization for ankle prostheses, we hypothesized that the optimized control parameters would result in better outcomes (e.g. reduced metabolic cost or increased preference) compared to any chosen generic parameter set.

In addition, we have now provided a supplementary figure demonstrating the metabolic cost trends over time, which allows readers to compare the initial and optimized conditions.

Comment R2-4: When comparing the results of the emulator (with optimized or generic control) with the prescribed foot. Did you correct for differences in mass of the prosthesis. And could additional differences between the emulator and prescribed prosthesis account for the higher energy cost with the emulator? For instance: roll over profile or alignment of the foot relative to the socket?

- Response: This is a good point to clarify. Although we did test the passive prosthesis, the primary comparison point that we were concerned with addressing in these studies was between the generic and optimized controllers, due to the fact that prosthesis mass and design was controlled. However, we agree that this is an important point to address in the discussion section, and have added the following sentence in the discussion section:

Both this lack of adaptation time and the differences in prosthesis design could account for the case study results demonstrating that metabolic cost was always lowest with the participants' passive prosthesis.

Comment R2-5: Energy cost is used as the cost function in optimization (and evaluation) . The authors refer to some limitation in this cost function. Limited reliability might be one (both limited reproducibility between trials as well as the error introduced in curve fitting beyond 2 min). Could the authors envisage the use of other cost functions in the future, potentially related to gait speed or stability?

- Response: We agree that metabolic cost measurements are noisy and that other cost functions related to gait speed or stability would be useful to examine for people with amputation. We have expanded on this point in the discussion section as follows:

However, people with amputation, compared to unaffected individuals, may have additional constraints affecting their gait. For example, individuals with amputation tend to fall more frequently [3] and list socket discomfort as a major limitation [39]. Perhaps because of the additional importance of stability and comfort, people with amputation adapt gait patterns with a similar energy cost independent of prosthesis behavior...Future work could examine alternate cost functions such as walking speed or stability.

Minor comments

Comment R2-6: page 5 line 31 : could the heel stiffness control condition also result in net positive work generation (negative damping) . Could you explain why this was not found in the optimized condition?

- Response: There was a typo in the explanation of the heel stiffness controller, as a positive damping coefficient (now labeled 'work constant' in response to Comment R1-2) corresponds with net positive work generation. Therefore, the optimized parameters do provide net positive work generation for the heel stiffness controller. The typo in the methods section of the heel stiffness controller has been corrected as follows:

The work constant is also used to distinguish which points corresponds to the loading and unloading phases. A positive value of the work constant indicates that work is injected during the gait cycle, so the heel produces more torque in the loading phase as the heel is pushing off than during the unloading phase. A negative work constant indicates that the heel produces less torque in the loading phase, resulting in net negative work over the gait cycle.

In addition, this point has also been clarified in the caption of Figure 1A:

Stiffness was varied between 50 and 130 Nm/rad, and the work constant varied from -0.5 to 0.5, with a larger positive work constant resulting in more positive work injected during the gait cycle.

Comment R2-7: Page 6, line 37: Am I correct to conclude that the optimal solution in the 5 parameter controller preferred the behaviour of a passive ankle spring, as peak active torque was negligible? Could you explain that finding?

- Response: Yes, it is correct that the optimal parameters in the 5-parameter controller resulted in behavior similar to a passive spring. Previous work has shown that injecting additional mechanical power using an active prosthesis does not necessarily correlate with reductions in metabolic rate [21, 22]. This example corroborates these findings, as the optimized controller, which essentially behaved as a passive spring, resulted in

slightly lower metabolic cost compared to the generic controller, which injected additional mechanical power. This point has been clarified in the results section:

Note that due to the small magnitude of the optimized peak torque, the optimized controller approximated the behavior of a passive spring, and therefore varying other timing parameters would have minimal effect on the resulting torque profile.

We have also added a sentence in the discussion addressing this point as well:

In fact, the largest metabolic reductions seen in these case studies compared an optimized controller that behaved as a passive spring to a generic controller that injected mechanical power into the gait cycle, corroborating previous evidence that providing higher mechanical power with an assistive device does not necessarily correlate with reduced user metabolic rate [21,22].

[21] Quesada, Caputo, & Collins. 2016. Increasing ankle push-off work with a powered prosthesis does not necessarily reduce metabolic rate for transtibial amputees. *Journal of Biomechanics* 49, 3452-3459.

[22] Gardinier et al. 2018. A controlled clinical trial of a clinically-tuned powered ankle prosthesis in people with transtibial amputation. *Clinical Rehabilitation*. 32, 319-329.

Comment R2-8: page 6, line 55: why did you not use similar generic control for reference in the 4 and 5 parameter optimization?

- Response: The generic control parameters were simply to have a point of comparison within the feasible range for each specific controller, which the optimized controller should have been able to outperform. For the time-based torque controller case studies, we chose the generic timing and magnitude control parameters within a range that was previously shown to be successful with exoskeletons, with modifications to these parameters and the stiffness parameter based on subject comfort and the stiffness of their prescribed prosthesis. We have added a sentence in the introduction to the time-based torque controller outlining our decisions in choosing generic control parameters:

In choosing the parameters for the generic controller, the stiffness parameter was based on the stiffness of the participant's prescribed prosthesis, while the remaining magnitude and timing parameters were based on average optimal parameters for ankle exoskeletons, with subject-specific modifications made for subject comfort.

Comment R2-9: page 7, line 50: on what did you base the parameters of the generic control condition in this case?

- Response: As mentioned in Comment R2-8, we add the following sentence to the manuscript describing the choice of all time-based torque control generic parameters:

In choosing the parameters for the generic controller, the stiffness parameter was based on the stiffness of the participant's prescribed prosthesis, while the remaining magnitude

and timing parameters were based on average optimal parameters for ankle exoskeletons, with subject-specific modifications made for subject comfort.

Comment R2-10: page 9, line 23: what do you mean by 'optimization seed'. Is that the initial condition of the optimization process

- Response: Yes, this is correct. This has been clarified in the text as follows:

In validation, in addition to comparing to the generic controller that emulated average optimal magnitude and timing parameters from ankle exoskeletons (stiffness = 900 Nm/rad, peak time = 80%, rise time = 20%, fall time = 15%), the initial condition used to seed the optimization was also tested (stiffness = 800 Nm/rad, peak time = 65%, rise time = 30%, fall time = 20%) (Fig. 4B).

Comment R2-11: page 9, line 35: avoid the word "significant change" here as it usually refers to statistical testing, which was not performed in this study. Use something like "consistent improvements"

- Response: Thank you for pointing this out; we have replaced the word "significant" with "functionally relevant".

Comment R2-12: page 10 line 28: Better convergence to an optimal solution does not only depend on the time of each condition (as in extended protocol) but also on the number of generations and number of iterations per generation I believe.

- Response: Yes, we completely agree. We decided not to continue testing additional generations because we were not seeing consistent improvements over time. We address this point in the discussion section:

Nevertheless, it is possible that better results might be achieved with additional training, continued optimization or other protocol improvements.

Comment R2-13: page 10, line 35: what do you mean by "metabolic cost variation"

- Response: This phrase would likely be more clearly described as "metabolic cost variability" and has been changed in the text. When human-in-the-loop optimization is successfully used for exoskeletons, parameters converge to tighter bounds during each generation, which then typically reduces the variability in metabolic cost. However, we did not see this trend in our case studies (See figure created in response to Comment R2-2).

Comment R2-14: Page 11, paragraph 2: I'm not convinced that the lack of the ankle (compared to exoskeleton users) provides less sensory feedback. It is unclear whether and how torque enhancement at the ankle is sensed by ankle proprioceptors or musclesensors. It might well be sensed in the interface between device and leg as in the socket of the prosthesis. Moreover, control might be easier when the device does not have to cooperate with the biological ankle muscle but has a one-to-one effect to ankle torque provided?

- Response: In response to this comment, as well as Comment R1-21, we have softened the claim that proprioception is inherently worse at the ankle for people with amputation, and instead discuss how the typical sensory pathways are disrupted as follows:

In contrast, proprioception in people with amputation is altered, since sensory information is not available directly from the limb. Although people with amputation can fairly precisely sense the stiffness of their prosthesis, sensory information must come from interactions at the socket and whole-body proprioception of resulting knee and hip kinematics [40].

Comment R2-15: page 11, line 42: prioritizing stability might also require extra metabolic energy. Hence better stability might provide lower metabolic cost?

- Response: We agree that prioritizing stability would likely require a higher metabolic cost and amend the discussion to explicitly address this as follows:

Or perhaps the "forced exploration" that comes from trying many diverse conditions, which is shown to be beneficial in pushing a person towards the metabolic minimum while walking with exoskeletons [42], is actually harmful to the learning process of people with amputation because "bad" conditions force the participant to prioritize stability over metabolic cost minimization, which could result in increased metabolic cost.

Comment R2-16: page 11, line 44: "There are a myriad of reasons why HILO may be more challenging when applied to powered prostheses instead of exoskeletons" maybe the biggest difference is not in the device but in the fact that these exoskeleton studies are generally performed in able-bodied persons?

- Response: We agree and address this statement in the discussion section as follows:

Independent from the optimization variables that can affect the results, it is possible that the contrast between the success of HILO for exoskeletons and the failure of HILO for active prostheses is related to the differences between participant groups.

Comment R2-17: Page 13, line 15: visualize or describe this optimization process, outline what is meant by generation, control law and provide an example of how energy cost converged to an optimum in this process.

- Response: We do not claim that energy cost converged to an optimum in these case studies, but we have added a supplemental figure visualizing the metabolic cost as a function of time over the generations. We have also added the following sentence to better explain the optimization process:

In the optimization protocol, a number of control laws are evaluated in each generation, after which the optimizer determines a new distribution from which to select the control laws for the following generation.

Having previously published more lengthy explanations of the algorithm, we did not think it appropriate to repeat those details here.

Comment R2-18: Page 13, line 45-48: define heel angle and describe how inversion-eversion is controlled in the 3DOF foot.

- Response: Thank you for pointing out that heel angle should be defined. We have revised the text to include the following description:

Heel angle is defined in the sagittal plane as the angle from the plane perpendicular to the prosthesis pylon to the heel end-effector, as described in [43].

In addition, we define the control scheme that dictates control of inversion-eversion as follows:

The two toes of the device were connected to passive compression springs, while the heel was the only actively controlled digit.

Comment R2-19: Page 14, line 43: explain the parameters in the balance controller (K and τ) in little more detail.

- Response: We agree that these parameters should be more thoroughly explained and have added the following:

Two parameters were optimized this control architecture, where the prosthesis behaved as a passive spring in plantarflexion and dorsiflexion, with a baseline nominal inversion/eversion torque parameter t_{nom} and a gain K dictating the magnitude of correction for center of mass velocity deviations.